# SEINT: An Efficient SE($p$)-Invariant Transport Metric Driven by Polar Transport Discrepancy-based Representation

**Junyi Lin**[*1], **Dunyao Xue**[*1], **Jun Yu**[2], **Hongteng Xu**[†3], **Cheng Meng**[†1, 4]

[1]Institute of Statistics and Big Data, Renmin University of China, Beijing, China
[2]School of Mathematics and Statistics, Beijing Institute of Technology, Beijing, China
[3]Gaoling School of Artificial Intelligence, Renmin University of China, Beijing, China
[4]Center for Applied Statistics, Renmin University of China, Beijing, China
`{junyilin, xuedunyao1202, chengmeng}@ruc.edu.cn,`
`yujunbeta@bit.edu.cn, hongtengxu@ruc.edu.cn`

## Abstract

We introduce SEINT, a novel Special Euclidean group-Invariant (SE($p$)) metric for comparing probability distributions on $p$-dimensional measured Banach spaces. Existing SE($p$)-invariant alignment methods often face high computational costs or lack metric guarantees. To overcome these limitations, we develop a polar transport discrepancy combined with distance convolution to extract SE($p$)-invariant representations. These representations are then used to compute the alignment between two distributions via optimal transport. Theoretically, we prove that SEINT is a well-defined metric on the space of isometry classes of normed vector spaces. Beyond its inherent SE($p$)-invariance, SEINT also supports cross-space distribution comparison. Computationally, SEINT aligns two samples of size $n$ with a complexity of just $\mathcal{O}(n \log n)$ to $\mathcal{O}(n^2)$. Extensive experiments validate its advantages: As a robust metric, it outperforms or matches existing SE($p$)-invariant methods in classification and cross-space tasks under isometries. As a regularizer, it greatly enhances molecular generation performance across both pre-training and fine-tuning tasks, achieving state-of-the-art (SOTA) results on key benchmarks. The code is available at `https://github.com/junyilin559/SEINT`.

## 1 Introduction

Optimal Transport (OT) and the Wasserstein distance offer a powerful mathematical framework for comparing probability distributions (Villani et al., 2008; Peyré et al., 2019). These tools have driven significant advances across a broad range of fields, including statistics (Meng et al., 2020; 2019; Deb et al., 2021; Li et al., 2023c; Zhang et al., 2023), machine learning (Courty et al., 2016; Laclau et al., 2017; Kolouri et al., 2018; Xu et al., 2022; Hu et al., 2025), computer vision (Tolstikhin et al., 2018; Arjovsky et al., 2017; Tong et al., 2024), natural language processing (Kusner et al., 2015; Zhao et al., 2019; Chen et al., 2023), and reinforcement learning (Metelli et al., 2019; Liu et al., 2020; He et al., 2022b). However, the Wasserstein distance has inherent limitations in addressing structural learning problems involving geometric invariance, such as (Ganea et al., 2022; Song et al., 2023; Bose et al., 2024; Yue et al., 2025), point cloud registration Shen et al. (2021); Yu et al. (2023), and multimodal alignment (Qiu et al., 2023; Alatkar & Wang, 2023). For objects with translation- and rotation-invariant shape properties (e.g., molecules or 3D point clouds), an effective metric must be invariant under the Special Euclidean group. Although the Wasserstein distance provides a general framework for distribution comparison, its standard formulation lacks native SE($p$)-invariance, requiring problem-specific adaptations to handle such geometric symmetries.

To the best of our knowledge, existing OT-based SE($p$)-invariant methods can be classified into three main strategies, as illustrated in Fig. 1.

---

[*]Authors contributed equally.
[†]Corresponding authors.

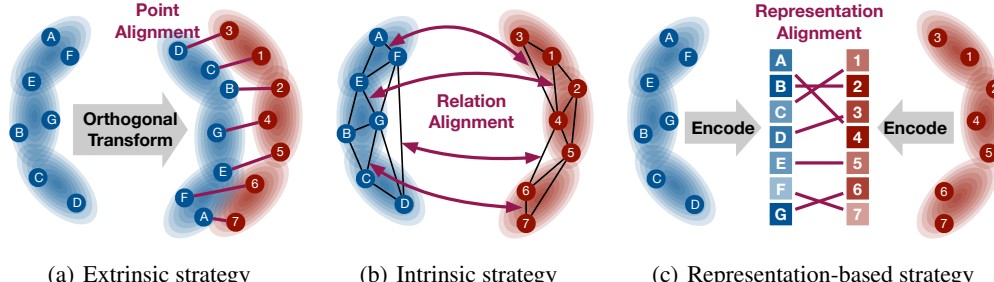

(a) Extrinsic strategy      (b) Intrinsic strategy      (c) Representation-based strategy

Figure 1: Overview of existing OT-based SE($p$)-invariant methods. (a) shows the *Extrinsic strategy* that jointly optimize the OT problem and the orthogonal transformation; (b) shows the *Intrinsic strategy* that leverages the geometric structure of data; (c) shows the *Representation-based strategy* that align data using SE($p$)-invariant features.

The first strategy *extrinsically* enforces SE($p$) invariance by jointly optimizing an orthogonal matrix in $\mathbb{R}^p$. Representative approaches include: EMD under Transformation Sets ($\mathrm{EMD}^{\mathcal{G}}$) (Cohen & Guibasm, 1999), which searches over a general transformation group $\mathcal{G}$, and Softassign Procrustes Matching (SPM) (Rangarajan et al., 1997), which specializes this paradigm to orthogonal transformations. More recent methods in this extrinsic family include the *Wasserstein Procrustes* formulations (Grave et al., 2019; Alvarez-Melis et al., 2019) and their extensions to non-Euclidean domains (Alaya et al., 2022; Beier et al., 2025). However, these iterative schemes often lead to significant computational overhead due to repeated OT calculations. To address efficiency concerns, methods like Rotational Invariant Sliced Gromov-Wasserstein (RISGW) (Vayer et al., 2019) utilize a closed-form solution by projecting data onto 1D subspaces. While this reduces computational complexity, it sacrifices the metric properties of the distance. The second strategy adopts an *intrinsic* approach by leveraging the geometric structure of data, e.g., Gromov–Hausdorff (GH) distance (Gromov, 1981) and Gromov–Wasserstein (GW) distance (Mémoli, 2011). This intrinsic strategy achieves SE($p$) invariance by comparing metric spaces up to isometry, offering two key advantages: natural incorporation of SE($p$) transformations and the ability to perform cross-space comparisons. However, this strategy usually comes with significantly higher computational complexity compared to extrinsic approaches. Although some GW variants, such as Quantized GW (Chowdhury et al., 2021) and Sampled GW (Kerdoncuff et al., 2021), provide fast approximations to the original GW distance, they may lose some of its desirable metric properties.

The third strategy employs a *representation-based* approach, which directly extracts SE($p$)-invariant features from the data. This category includes classical methods such as Spherical Harmonic Representations (SHR) (Kazhdan et al., 2003) and statistical descriptors based on signatures, histograms, or transformation-derived features (Tombari et al., 2010; Guo et al., 2013; Rusu et al., 2009; Beckmann et al., 2025; Piening & Beinert, 2025), as well as modern learning-based approaches such as neural networks (Yew & Lee, 2020; Saleh et al., 2022; Wang et al., 2022) and Rotation-Invariant

Table 1: Properties of OT-based SE($p$)-invariant methods.

| Type | Method | Complexity | Metric | Isometry |
|---|---|---|---|---|
| Extrinsic | $\mathrm{EMD}^{\mathcal{G}}$ | $\mathcal{O}(n^3)$-$\mathcal{O}(n^3 \log n)$ | ✓ | ✗ |
| | SPM | $\mathcal{O}(n^2 \log n)$ | ✗ | ✗ |
| | RISGW | $\mathcal{O}(n \log n)$ | ✗ | ✓ |
| Intrinsic | GH | NP-hard | ✓ | ✓ |
| | GW | $\mathcal{O}(n^3)$-$\mathcal{O}(n^4)$ | ✓ | ✓ |
| Representation | SHR | - | ✗ | ✗ |
| | RIT | - | ✗ | ✗ |
| | **SEINT**(Ours) | $\mathcal{O}(n \log n)$-$\mathcal{O}(n^2)$ | ✓ | ✓ |

[1] **Complexity** here is evaluated under the assumption $n \gg p$.
[2] **Metric** indicates whether the method is a well-defined metric.
[3] **Isometry** indicates whether the method's applicability extends to comparing general isometry classes.

Transformers (RIT) (Yu et al., 2023). While these methods demonstrate strong empirical performance, they may induce a pseudometric rather than a true metric, since the representation process can potentially discard some geometric information.

The comparative analysis in Table 1 reveals that existing strategies - whether extrinsic, intrinsic, or representation-based - fail to simultaneously optimize for computational efficiency, rigorous metric

properties, and general applicability across isometry classes. To address these limitations, we propose the **SE(*p*)-Invariant Transport (SEINT)** metric, which aims to achieve this crucial balance.

Our contributions can be summarized as follows:

1) **Novel Representation Route:** We introduce two innovative unsupervised representation techniques, called Polar Transport Discrepancy (PTD) and Distance-convoluted Polar Transport Discrepancy (DcPTD), which generate 1D SE(*p*)-invariant features without training. These two techniques form the foundation for the SEINT distance through optimal transport.
2) **Strong Metric Foundation:** We prove that SEINT is a proper metric on isometry classes of normed spaces, which inherently guarantees SE(*p*)-invariance for data embedded in the normed space. In addition, SEINT also enables cross-space comparison of distributions.
3) **Efficient Numerical Implementation:** The SEINT metric achieves $\mathcal{O}(n^2)$ time complexity in general cases. For scenarios where the ground distance matrix is decomposable, we demonstrate that this complexity can be further reduced to $\mathcal{O}(n \log(n))$, enabling large-scale applications.

Experimental results show that SEINT performs well on point cloud classification and cross-space tasks, and it also works effectively as a regularization term. For molecule generation, SEINT brings substantial improvements to baseline methods and achieves state-of-the-art (SOTA) performance on key metrics. These results validate SEINT as a practical SE(*p*)-invariant metric for structured data and an effective tool for regularizing complex generative models.

## 2 PROPOSED METHOD

### 2.1 MOTIVATION AND BRIEF PRINCIPLE

Developing effective SE(*p*)-invariant metrics is crucial for solving structural learning problems. We address this challenge through an optimal transport-based framework that utilizes novel SE(*p*)-invariant features constructed from measures in the *measured Banach space*.

Throughout this paper, we denote a measured Banach space via the triplet $(X, \|\cdot\|_X, \mu_X)$. Here, $(X, \|\cdot\|_X)$ is a Banach space (a complete, separable normed vector space), $\mathcal{B}(X)$ is its Borel $\sigma$-algebra, and $\mu_X$ be a probability measure on the measurable space $(X, \mathcal{B}(X))$[1]. Consider two measured Banach spaces $(X, \|\cdot\|_X, \mu_X)$ and $(Y, \|\cdot\|_Y, \mu_Y)$ with their induced metrics $d_X$ and $d_Y$. Without loss of generality, we assume the distributions under consideration are centered. Furthermore, we assume that any relevant isomorphism maps the base point $0$ to the base point $0$. These assumptions effectively standardize the spaces with respect to translation, meaning we disregard differences arising purely from translational shifts.

Drawing inspiration from optimal transport in polar coordinates, we introduce two novel unsupervised representation techniques: the Polar Transport Discrepancy (PTD) and its enhanced variant, the Distance-convoluted Polar Transport Discrepancy (DcPTD). These training-free techniques effectively extract SE(*p*)-invariant features from measures in Banach spaces. We now provide formal definitions of PTD and DcPTD, followed by the proposed SEINT approach.

### 2.2 POLAR TRANSPORT DISCREPANCY AND DISTANCE-CONVOLUTED POLAR TRANSPORT DISCREPANCY

Given two measured Banach spaces $(X, \|\cdot\|_X, \mu_X)$ and $(Y, \|\cdot\|_Y, \mu_Y)$ with induced distance $d_X$ and $d_Y$, a probability measure $\pi$ on their product space $X \times Y$ is a *coupling* of $\mu_X$ and $\mu_Y$ if its marginals satisfy $(\text{proj}_X)_{\#}\pi = \mu_X$ and $(\text{proj}_Y)_{\#}\pi = \mu_Y$, where $\text{proj}_X$ and $\text{proj}_Y$ are the canonical projections. The set of all such couplings is denoted by $\Pi(\mu_X, \mu_Y)$. For a measurable map $f : X \to Y$, the *push-forward measure* $f_{\#}\mu_X$ on $(Y, \mathcal{B}(Y))$ is defined by $f_{\#}\mu_X(A) := \mu_X(f^{-1}(A))$ for any set $A \in \mathcal{B}(Y)$. If $f$ is a surjection satisfying $d_Y(f(x), f(x')) = d_X(x, x')$ for all $x, x' \in X$, it is called an *isometry* between $X$ and $Y$. If such an isometry exists, the two spaces are said to be *isometric*[2]. Consider a reference probability measure $\mu_Z \in \mathcal{P}(\mathbb{R})$ on the 1D Banach space

---

[1]Since a norm induces a metric via $d_X(x, x') := \|x - x'\|_X$, it follows that any measured Banach space $(X, \|\cdot\|_X, \mu_X)$ is also a metric measure space $(X, d_X, \mu_X)$ with this induced metric.

[2]Note that, if the spaces $(X, \|\cdot\|_X, \mu_X)$ and $(Y, \|\cdot\|_Y, \mu_Y)$ are isometric via such an isometry, their push-forward measure of the norm must coincide, i.e., $(\|\cdot\|_X)_{\#}\mu_X = (\|\cdot\|_Y)_{\#}\mu_Y$.

$(\mathbb{R}, |\cdot|)$, where $\mathcal{P}(\mathbb{R})$ denotes the space of probability measures on $\mathbb{R}$. We define a cost function $c : X \times \mathbb{R} \to \mathbb{R}_{\geq 0}$ by $c(x, z) = |\|x\|_X - z|$. This is equivalent to computing the optimal transport between the distribution of $\|X\|_X$ and $\mu_Z$, and hence the existence of an optimal coupling is guaranteed. We denote the set of all such optimal couplings by:

$$\Pi^*_{X,Z} := \left\{ \pi \in \Pi(\mu_X, \mu_Z) : \mathbb{E}_\pi |\|x\|_X - z| = \inf_{\pi' \in \Pi(\mu_X, \mu_Z)} \mathbb{E}_{\pi'} |\|x\|_X - z| \right\}. \tag{1}$$

For any $\pi^*_{X,Z} \in \Pi^*_{X,Z}$, we can disintegrate it as $d\pi^*_{X,Z}(x, z) = d\pi^*_{Z|X=x}(z)d\mu_X(x)$, where $\pi^*_{Z|X=x}$ is the conditional probability measure on $\mathbb{R}$ given $x \in X$. We define the *Polar transport discrepancy* as follows.

**Definition 1 (Polar Transport Discrepancy (PTD))** *Given a measured Banach space $(X, \|\cdot\|_X, \mu_X)$, a reference distribution $\mu_Z \in \mathcal{P}(\mathbb{R})$, and an optimal coupling $\pi^*_{X,Z} \in \Pi^*_{X,Z}$ from equation 1, the* Polar Transport Discrepancy $\zeta_{\pi^*_{X,Z}} : X \to \mathbb{R}_{\geq 0}$ *is defined as:*

$$\zeta_{\pi^*_{X,Z}}(x) := \int_{\mathbb{R}} |\|x\|_X - z| \, d\pi^*_{Z|X=x}(z). \tag{2}$$

Note that PTD exhibits a key property that enhances its flexibility. Specifically, the value of PTD depends on the selection of both the reference distribution $\mu_Z$ and the particular optimal coupling $\pi^*_{X,Z}$ (in cases of non-unique optimal couplings). This dependency enables the generation of multiple distinct feature representations, making PTD particularly versatile for capturing different aspects of the measure $\mu_X$.

However, one key question remains: Do PTD-derived features suffice for building a true isometry-invariant metric between two measures, rather than just a pseudometric? Existing literature suggests this may not hold, as intrinsic distance information proves indispensable for constructing proper isometry-invariant metrics, as demonstrated in foundational works on Gromov-type distances (Mémoli, 2011; Sturm, 2023). To address this limitation, we develop an enhanced approach that synergistically combines PTD with intrinsic distance information through a kernel-like transformation. This leads to our novel isometry-invariant feature extraction function, formally defined as follows.

**Definition 2 (Distance-convoluted Polar Transport Discrepancy (DcPTD))** *Given a measured Banach space $(X, \|\cdot\|_X, \mu_X)$ with induced metric $d_X$, a reference measure $\mu_Z \in \mathcal{P}(\mathbb{R})$, an optimal coupling $\pi^*_{X,Z} \in \Pi^*_{X,Z}$ from equation 1, and the corresponding polar transport discrepancy $\zeta_{\pi^*_{X,Z}}$, the* Distance-convoluted Polar Transport Discrepancy $\phi_{\pi^*_{X,Z}} : X \to \mathbb{R}_{\geq 0}$ *is defined as:*

$$\phi_{\pi^*_{X,Z}}(x) := \int_X d_X(x, x')\zeta_{\pi^*_{X,Z}}(x') \, d\mu_X(x'). \tag{3}$$

**Remark 1** *DcPTD exhibits two key properties that ensure its soundness as a representation. (1) It satisfies rigorous **isometric consistency**: for any isometry $f : X \to Y$, we have $\phi_{\pi^*_{X,Z}}(x) = \phi_{\pi^*_{Y,Z}}(f(x))$, thus guaranteeing the invariance of extracted representations under rigid isometric transformations. (2) It yields a **dimension-independent** representation, as the output $\phi_{\pi^*_{X,Z}}(x)$ is always a non-negative scalar value on the real line space $(\mathbb{R}, |\cdot|)$, regardless of the ambient dimension of the input space.*

These two unique properties of DcPTD enable a novel approach for comparing distributions across different spaces. Given two measured Banach spaces $(X, \|\cdot\|_X, \mu_X)$ and $(Y, \|\cdot\|_Y, \mu_Y)$ with induced metrics $d_X$ and $d_Y$, we leverage DcPTD's isometry-invariant, dimension-independent characteristics through the mappings $(\phi_{\pi^*_{X,Z}}, \phi_{\pi^*_{Y,Z}})$ to encode structural information of two input distributions $\mu_X$ and $\mu_Y$ into a common representation domain. The encoding step transforms the cross-space comparison problem into a tractable optimal transport computation in the same space, i.e., by computing

$$\inf_{\pi \in \Pi(\mu_X, \mu_Y)} \left( \mathbb{E}_\pi [|\phi_{\pi^*_{X,Z}}(x) - \phi_{\pi^*_{Y,Z}}(y)|^p] \right)^{\frac{1}{p}}. \tag{4}$$

This Optimal Transport computation capitalizes on DcPTD's ability to maintain geometric invariance while providing a unified framework for distribution comparison across potentially heterogeneous spaces. One key question that remains is how to select the reference measure $\mu_Z$. We now address this by establishing principled criteria for $\mu_Z$ selection, followed by presenting the SEINT metric definition.

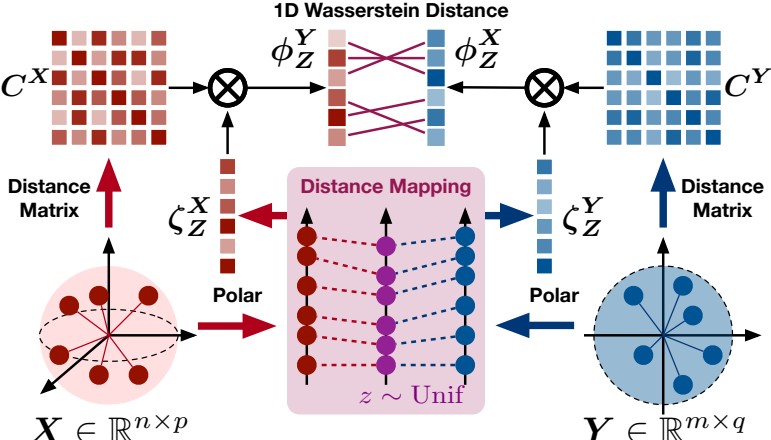

Figure 2: An illustration of the numerical implementation of SEINT. For the input data from two spaces, $\mathbf{X}$ and $\mathbf{Y}$, their respective norms are first calculated. Then, for each norm, a PTD is computed through a randomly generated reference $z$. PTD is then convolved with a distance matrix to obtain a DcPTD. Finally, the 1D Wasserstein distance between the DcPTDs yields the corresponding distance.

## 2.3 SPECIAL EUCLIDEAN GROUP-INVARIANT TRANSPORT (SEINT) METRIC

To simplify the choice of optimal coupling $\pi^*_{X,Z}, \pi^*_{Y,Z}$ in 1, we restrict the selection of $\mu_Z$ and make them directly connect to the data measures $\mu_X$ and $\mu_Y$. Firstly, we construct the candidate set of $Z$ as the family of distributions conditioned on $X$, i.e.,

$$\mathcal{P}_X(\mathbb{R}) = \left\{ \mu_{Z|X} \in \{X \to \mathcal{P}(\mathbb{R})\} : \exists \pi^*_{X,Z} \in \Pi^*_{X,Z} \text{ s.t. } \pi^*_{X,Z} = \mu_X \mu_{Z|X} \right\}. \tag{5}$$

This guarantees that every $\mu_Z$ is associated with a corresponding optimal coupling $\pi^*_{X,Z} \in \Pi^*_{X,Z}$. Given a reference measure $\mu_Z \in \mathcal{P}_X(\mathbb{R})$ and its associated optimal coupling $\pi^*_{X,Z}$, for any coupling $\pi \in \Pi(\mu_X, \mu_Y)$, the optimal coupling on the other side $(\pi^*_{Y,Z} \in \Pi^*_{Y,Z})$ must satisfy

$$\pi^*_{Y,Z} \in \text{argmin}_{\pi^{*\prime}_{Y,Z} \in \Pi^*_{Y,Z}} \mathbb{E}_\pi \left( |\zeta_{\pi^*_{X,Z}}(x) - \zeta_{\pi^{*\prime}_{Y,Z}}(y)| \right), \tag{6}$$

where $\zeta_{\pi^*_{X,Z}}$ and $\zeta_{\pi^{*\prime}_{Y,Z}}$ are the PTDs in Definition 1. By restricting $\mu_Z$ to $\mathcal{P}_X$ and adding condition 6, we ensure that the other optimal coupling $\pi^*_{Y,Z}$ is also associated.

Similarly, we define $\mathcal{P}_Y(\mathbb{R})$, and consider their union $\mathcal{P}_{X,Y}(\mathbb{R}) = \mathcal{P}_X(\mathbb{R}) \cup \mathcal{P}_Y(\mathbb{R})$, where condition 6 applies symmetrically on the opposite side. With this construction, selecting any reference measure $\mu_Z$ specifies a pair of optimal couplings $(\pi^*_{X,Z}, \pi^*_{Y,Z})$. Thus, the optimization reduces to searching over $\mu_Z$ within $\mathcal{P}_{X,Y}(\mathbb{R})$, and identifying a single least favorable reference measure $\mu_{Z_0}$ is adequate for comparing the distributions on the two metric spaces. The distance obtained under this formulation defines our proposed SEINT metric, formally stated as follows.

**Definition 3 (SE($p$)-Invariant Transport (SEINT) Distance)** *Consider two measured Banach spaces $(X, \|\cdot\|_X, \mu_X)$ and $(Y, \|\cdot\|_Y, \mu_Y)$ with induced metrics $d_X$ and $d_Y$. Let $\mathcal{P}_{X,Y}(\mathbb{R}) = \mathcal{P}_X(\mathbb{R}) \cup \mathcal{P}_Y(\mathbb{R})$ denoted in equation 5. For a given reference measure $\mu_Z \in \mathcal{P}_{X,Y}$, let $\pi^*_{X,Z} \in \Pi^*_{X,Z}$ and $\pi^*_{Y,Z} \in \Pi^*_{Y,Z}$ be the corresponding optimal couplings. Let $\phi_{\pi^*_{X,Z}}$ and $\phi_{\pi^*_{Y,Z}}$ be the DcPTD maps associated with couplings $\pi^*_{X,Z}$ and $\pi^*_{Y,Z}$. The SEINT distance is defined by finding an optimal coupling $\pi \in \Pi(\mu_X, \mu_Y)$ that minimizes the least favorable measure $\mu_Z$ over all choices $\mathcal{P}_{X,Y}(\mathbb{R})$:*

$$\mathcal{L}_{\text{SEINT}}(X, Y, \mu_X, \mu_Y) := \inf_{\pi \in \Pi(\mu_X, \mu_Y)} \sup_{\mu_Z \in \mathcal{P}_{X,Y}(\mathbb{R})} \left( \mathbb{E}_\pi \left[ |\phi_{\pi^*_{X,Z}}(x) - \phi_{\pi^*_{Y,Z}}(y)|^p \right] \right)^{\frac{1}{p}}. \tag{7}$$

The pipeline for calculating SEINT is outlined in Fig. 2. We now provide the following theorem, which demonstrates that SEINT is a well-defined distance metric and provides a characterization of isometry across different spaces.

**Theorem 1 (Metric Property on Isometry Classes)** $\mathcal{L}_{\text{SEINT}}$ *defines a metric on the space of isometry classes of metric measure spaces. Specifically, for any measured Banach spaces* $(X, \|\cdot\|_X, \mu_X)$, $(Y, \|\cdot\|_Y, \mu_Y)$ *and* $(Z, \|\cdot\|_Z, \mu_Z)$:

*1) **Identity of Indiscernibles:*** $\mathcal{L}_{\text{SEINT}}(X, Y, \mu_X, \mu_Y) \geq 0$, *and* $\mathcal{L}_{\text{SEINT}}(X, Y, \mu_X, \mu_Y) = 0$ *if and only if* $(X, d_X, \mu_X)$ *and* $(Y, d_Y, \mu_Y)$ *are isometric.*

*2) **Symmetry:*** $\mathcal{L}_{\text{SEINT}}(X, Y, \mu_X, \mu_Y) = \mathcal{L}_{\text{SEINT}}(Y, X, \mu_Y, \mu_X)$.

*3) **Triangle Inequality:*** $\mathcal{L}_{\text{SEINT}}(X, Y, \mu_X, \mu_Y) \leq \mathcal{L}_{\text{SEINT}}(X, Z, \mu_X, \mu_Z) + \mathcal{L}_{\text{SEINT}}(Y, Z, \mu_Y, \mu_Z)$.

The proof of the theorem is detailed in the Appendix B. Notably, SEINT extends naturally to arbitrary metric measure spaces equipped with a well-defined base point; see the Appendix B.3 for additional theoretical extensions and related results. In addition, recall that the Special Euclidean group SE($p$) represents the orientation-preserving isometries (rigid transformations) of $\mathbb{R}^p$. Given SEINT's explicit design for invariance under such transformations, we immediately obtain the following corollary regarding its invariance property.

**Corollary 1 (SE($p$) Invariance)** *Let* $(X, \|\cdot\|_X, \mu_X)$ *and* $(Y, \|\cdot\|_Y, \mu_Y)$ *be two measured Banach spaces embedded in Euclidean spaces, where* $X \subseteq \mathbb{R}^p$. *For any transformation* $g = (R, t) \in$ SE($p$) $:= \text{SO}(p) \ltimes \mathbb{R}^p$, *its action on $X$ is defined as* $g(X) := RX + t$. *Define the transformed space as* $(g(X), \|\cdot\|_{g(X)}, g_\# \mu_X)$. *Then SEINT is invariant under the* SE($p$) *transformations, i.e.,*

$$\mathcal{L}_{\text{SEINT}}(X, Y, \mu_X, \mu_Y) = \mathcal{L}_{\text{SEINT}}(g(X), Y, g_\# \mu_X, \mu_Y). \tag{8}$$

## 2.4 NUMERICAL IMPLEMENTATION

In the discrete real-valued setting, let $\mathbf{X} = (\boldsymbol{x}_1, \ldots, \boldsymbol{x}_n)^\top \in \mathbb{R}^{n \times p}$ be a sample matrix. The associated empirical measure is $\mu_{\mathbf{X}} = \sum_{i=1}^n p_{\boldsymbol{x}_i} \delta_{\boldsymbol{x}_i}$, where $\boldsymbol{p}_{\mathbf{X}} = (p_{\boldsymbol{x}_1}, \ldots, p_{\boldsymbol{x}_n})^\top \in \Delta^{n-1}$ is a probability vector in the standard $(n-1)$-simplex. To enable efficient optimization of equation 7 in the discrete setting, we require only the following constraint on the space of reference measures, which leads to the corollary presented below.

**Corollary 2 (Metric Property for Discrete Distributions)** *Let two discrete measures be defined by their supports* $\mathbf{X} \in \mathbb{R}^{n \times p}$ *and* $\mathbf{Y} \in \mathbb{R}^{m \times q}$, *with corresponding probability vectors* $\boldsymbol{p}_{\mathbf{X}}$ *and* $\boldsymbol{p}_{\mathbf{Y}}$. *And their norms are bounded by* $M$. *Further, consider the space of discrete measures* $\hat{\mathcal{P}}(\mathbb{R}) = \{\mu_{\mathbf{Z}} = \sum_{j=1}^k \frac{1}{k} \delta_{z_j} : \{z_j\}_{1 \leq j \leq k} \in [0, M]^k\}$. *Then, under the assumption that values of the norm* $\|\boldsymbol{x}_i\|$ *($i = 1, \ldots, n$) are distinct and $k$ is large enough, SEINT defines a metric on this space of measures.*

The proof is detailed in Appendix B.4. Note that the *distinctness* assumption is generally satisfied in empirical distributions. Consider a reference distribution $\mu_Z$ uniformly drawn from $\hat{\mathcal{P}}(\mathbb{R})$ in Corollary 2. Under this distinctness condition, the set of optimal transport plans $\Pi^*_{X,Z}$ in equation 1 contains a unique element, which we denote $\Pi^\star \in \mathbb{R}^{n \times n}$. In this scenario, the values of the PTD from equation 2 depend only on $\mathbf{X}$ and $\mathbf{Z}$ and can be expressed as:

$$\boldsymbol{\zeta}^{\mathbf{X}}_{\mathbf{Z}_i} = \frac{1}{p_{\boldsymbol{x}_i}} \sum_{j=1}^k \Pi^\star_{ij} |\|\boldsymbol{x}_i\|_X - z_j|, \quad i = 1, \ldots, n. \tag{9}$$

The algorithm for the numerical implementation of SEINT is given below.

**Complexity Analysis.** The time complexity of Algorithm 1 involves several steps. Computing the distance matrices $\mathbf{C}^{\mathbf{X}}$ and $\mathbf{C}^{\mathbf{Y}}$ takes $\mathcal{O}(n^2 + m^2)$ time. According to the North-West corner rule(Peyré et al., 2019), computing $\boldsymbol{\zeta}^{\mathbf{X}}_{\mathbf{Z}_i}$ and $\boldsymbol{\zeta}^{\mathbf{Y}}_{\mathbf{Z}_i}$ takes $\mathcal{O}(n \log n + m \log m + k \log k)$ time. The matrix-vector multiplications $\mathbf{C}^{\mathbf{X}} \boldsymbol{\zeta}^{\mathbf{X}}_{\mathbf{Z}_i}$ and $\mathbf{C}^{\mathbf{Y}} \boldsymbol{\zeta}^{\mathbf{Y}}_{\mathbf{Z}_i}$ require $\mathcal{O}(n^2 + m^2)$ time. Computing the final distance between feature vectors requires $\mathcal{O}(n \log n + m \log m)$ time. Assuming $a$ is constant, the overall time complexity is thus $\mathcal{O}(n^2 + m^2)$. Note that the computational complexity of Algorithm 1 primarily stems from two operations: (1) distance matrix computation and (2) subsequent matrix-vector multiplications. For structured distance functions, e.g., squared Euclidean distances $d(x, y) = \|x - y\|_2^2$, we can exploit matrix decomposition techniques to accelerate the overall computation, resulting in an overall complexity of $\mathcal{O}(n \log n + m \log m)$ when $n \gg p, q$. Implementation details and accelerated algorithm are provided in the Appendix C and Algorithm 2.

---

**Algorithm 1** SEINT Computation

---

**Require:** Two datasets $\mathbf{X} \in \mathbb{R}^{n \times p}$ and $\mathbf{Y} \in \mathbb{R}^{m \times q}$, probability vectors $\boldsymbol{p_X} \in \Delta^{n-1}, \boldsymbol{p_Y} \in \Delta^{m-1}$, number of reference distributions $a$, support size for reference distributions $k$
 1: Initialize the data $\mathbf{X}$ and $\mathbf{Y}$.
 2: Generate $a$ random reference distributions $\mu_{\mathbf{Z}_1}, \mu_{\mathbf{Z}_2}, \ldots, \mu_{\mathbf{Z}_a}$ with $k$ samples.
 3: Calculate the distance matrix: $\mathbf{C^X} \leftarrow (\|\boldsymbol{x}_i - \boldsymbol{x}_j\|_X)_{ij}; \mathbf{C^Y} \leftarrow (\|\boldsymbol{y}_i - \boldsymbol{y}_j\|_Y)_{ij}$ $\triangleright \mathcal{O}(n^2 + m^2)$
 4: **for** $i \leftarrow 1$ to $a$ **do**
 5:     Compute $\boldsymbol{\zeta}_{\mathbf{Z}_i}^{\mathbf{X}}$ using $\mathbf{X}, \boldsymbol{p_X}, \mu_{\mathbf{Z}_i}$ via equation 9             $\triangleright \mathcal{O}(n \log n + k \log k)$
 6:     Compute $\boldsymbol{\zeta}_{\mathbf{Z}_i}^{\mathbf{Y}}$ using $\mathbf{Y}, \boldsymbol{p_Y}, \mu_{\mathbf{Z}_i}$ via equation 9            $\triangleright \mathcal{O}(m \log m + k \log k)$
 7:     $\boldsymbol{\phi}_{\mathbf{Z}_i}^{\mathbf{X}} \leftarrow \mathbf{C^X} \boldsymbol{\zeta}_{\mathbf{Z}_i}^{\mathbf{X}}; \boldsymbol{\phi}_{\mathbf{Z}_i}^{\mathbf{Y}} \leftarrow \mathbf{C^Y} \boldsymbol{\zeta}_{\mathbf{Z}_i}^{\mathbf{Y}}$                     $\triangleright \mathcal{O}(n^2 + m^2)$
 8:     Calculate the 1D-$\mathcal{W}_2$ loss: $\mathcal{L}_i \leftarrow \mathcal{W}_2 \left( (\boldsymbol{\phi}_{\mathbf{Z}_i}^{\mathbf{X}})_{\#} \mu_{\mathbf{X}}, (\boldsymbol{\phi}_{\mathbf{Z}_i}^{\mathbf{Y}})_{\#} \mu_{\mathbf{Y}} \right)$   $\triangleright \mathcal{O}(n \log n + m \log m)$
 9: **end for**
10: Set the final loss $\mathcal{L}_{\text{SEINT}} \leftarrow \max_{1 \le i \le a} \mathcal{L}_i$
11: **return** $\mathcal{L}_{\text{SEINT}}$

---

## 2.5 Integral-SEINT

SW distance (Bonneel et al., 2015) and Max-SW distance (Deshpande et al., 2019) are two popular variants of the standard Wasserstein distance. The key difference between them is that, SW integrates 1D Wasserstein distances with respect to all random projections, while Max-SW selects the projection that leads to the largest 1D Wasserstein distance. Drawing parallels to our SEINT framework, which adopts a worst-case perspective by maximizing over reference measures, we can similarly construct an integrated variant called Integral-SEINT (ISEINT):

$$\mathcal{L}_{\text{ISEINT}}(X, Y, \mu_X, \mu_Y) := \inf_{\pi \in \Pi(\mu_X, \mu_Y)} \left( \mathbb{E}_{\pi \times \mathcal{D}(\mathcal{P}_{X,Y}(\mathbb{R}))} \left[ |\phi_{\pi_{X,Z}^*}(x) - \phi_{\pi_{Y,Z}^*}(y)|^p \right] \right)^{\frac{1}{p}}, \quad (10)$$

where the expectation is taken over a suitable distribution $\mathcal{D}(\mathcal{P}_{X,Y}(\mathbb{R}))$ of 1D reference measures $\mu_Z \in \mathcal{P}_{X,Y}(\mathbb{R})$. A numerical approximation for ISEINT can be obtained by simply modifying Line 10 in Algorithm 1: we compute the average value instead of taking the maximum value.

## 3 Experiments

We conduct a systematic investigation of our method from two key perspectives: (1) a comprehensive analysis of its metric properties, and (2) a practical assessment of its performance as a regularizer in downstream applications. All experiments were conducted on a server with 256 GB RAM and 64 cores Intel ® Xeon ® Gold 5218 CPU, and a standard workstation equipped with a single NVIDIA RTX 4090 GPU. **Representative experiments are shown below, and more implementation details and results are in the Supplementary Material.**

## 3.1 As a distance metric

**SE($p$)-Invariance.** To empirically validate the SE($p$)-invariance property established in Corollary 1, we conduct experiments on 3D point cloud classification using a specially constructed ModelNet40-SE(3) dataset. This benchmark is derived from ModelNet40 (Wu et al., 2015) through the following augmentation protocol: (1) We randomly select an object from the original pre-aligned 3D models in each category. (2) Each base model undergoes 25 random SE(3) transformations (rigid rotations and translations); (3) We further add random Gaussian noise to models to increase variability. The final dataset contains 1000 transformed models. We selected eight SE($p$)-invariant metrics for comparison, including the GW distance (Mémoli, 2011), the Entropic-GW (EGW) distance (Peyré et al., 2016), the Rotation-Invariant Sliced GW (RISGW) distance (Vayer et al., 2019), SW distance (Bonneel et al., 2015), SGW distance (Vayer et al., 2019), and Wasserstein distance ($W_2$)(Cuturi, 2013). Notably, the first four metrics explicitly claim to possess rotation invariance. The regularization parameter for EGW is set to 0.1. The number of projections for RISGW, SW, and SGW, as well as the number of reference distributions for SEINT, are all set to 50.

As shown in Table 2, our method consistently achieves 100% accuracy, confirming that SEINT is a strictly SE($p$)-invariant metric. Moreover, SEINT attains the highest computational efficiency among

Table 2: Comparisons of classification accuracy (%) across different numbers of neighbors (k)

| Method | $\text{Acc}_{(k=1)}$ | $\text{Acc}_{(k=5)}$ | $\text{Acc}_{(k=10)}$ | Time (h) | SE($p$)-Inv |
|--------|------|------|------|------|------|
| GW | — | — | — | 2655.9 | ✓ |
| EGW | — | — | — | 2644.3 | ✓ |
| RISGW | $74.9_{\pm 3.9}$ | $59.5_{\pm 4.0}$ | $50.0_{\pm 3.6}$ | 400.6 | ✓ |
| $W_2$ | $52.5_{\pm 3.2}$ | $38.4_{\pm 2.3}$ | $31.3_{\pm 1.4}$ | 113.4 | ✗ |
| SGW | $59.1_{\pm 2.9}$ | $46.2_{\pm 4.2}$ | $40.9_{\pm 3.6}$ | 20.2 | ✗ |
| SW | $56.7_{\pm 4.1}$ | $41.6_{\pm 4.1}$ | $37.2_{\pm 4.0}$ | 16.6 | ✗ |
| **SEINT** | $\textbf{100.0}_{\pm \textbf{0.0}}$ | $\textbf{100.0}_{\pm \textbf{0.0}}$ | $\textbf{100.0}_{\pm \textbf{0.0}}$ | 9.01 | ✓ |
| **ISEINT** | $\textbf{100.0}_{\pm \textbf{0.0}}$ | $\textbf{100.0}_{\pm \textbf{0.0}}$ | $\textbf{100.0}_{\pm \textbf{0.0}}$ | **8.29** | ✓ |

[1] All reported times are CPU times.
[2] For GW/EGW, entries marked "—" indicate results that are not reported due to their prohibitive computational cost.

all evaluated approaches: it is faster than competing methods while uniquely preserving perfect accuracy, whereas alternatives such as SW and SGW remain slower and fall below $60\%$ accuracy. These results highlight SEINT's strong balance between efficiency and accuracy.

**Cross-Space Capability.** To evaluate the inter-space capability of our metric, we employ the horse-gallop dataset (Sumner & Popović, 2004), which contains a reference horse model and its dynamic galloping sequence. Our experimental protocol involves two steps: (1) Projecting the reference point cloud onto three orthogonal principal planes (XY, XZ, YZ) to create 2D variants, (2) Computing distances between each projected/reference space and all gallop-sequence point clouds.

The results in Fig. 3 reveal three key findings about SEINT's cross-space performance: (1) It precisely captures the gallop dynamics through three peaks that align with the horse's stride phases, outperforming EGW in terms of variation amplitude; (2) While GW and RISGW show larger fluctuations, their patterns deviate from the expected biomechanical sequence; (3) Most importantly, SEINT demonstrates strict metric consistency - its loss values increase monotonically as the source-target distribution divergence grows, validating its theoretical properties even in cross-space comparisons. This systematic evaluation demonstrates SEINT's ability to compare distributions across heterogeneous ambient spaces (2D vs 3D) while maintaining meaningful geometric relationships.

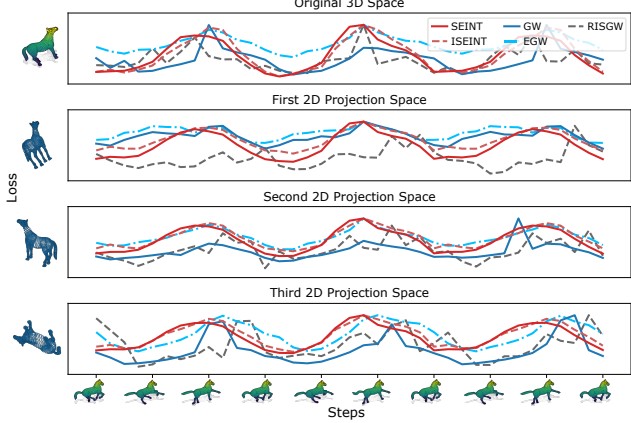

Figure 3: Loss evolution between the dynamic horse sequence and four references. The SEINT distance exhibits three distinct and consecutive peaks, while EGW shows similar qualitative trends but with a weaker response magnitude, and GW/RISGW fail to maintain consistent distance relationships.

**Capability on High-Dimensional Data.** We compare SEINT with several OT-based methods, including GW, Rotation-Invariant Sliced GW (RISGW), Sliced GW, and Wasserstein Distance, under high-dimensional settings. Let $\{\boldsymbol{x}_i\}_{i=1}^n$ and $\{\boldsymbol{y}_i\}_{i=1}^n$ be i.i.d. samples generated from $\mathcal{N}_d(\boldsymbol{0}_d, \boldsymbol{\Sigma}_X)$ and $\mathcal{N}_d(\boldsymbol{0}_d, \boldsymbol{\Sigma}_Y)$, respectively. We set $n = 200, d = 50$, $\boldsymbol{\Sigma}_X = \text{diag}(3\boldsymbol{I}_2, \boldsymbol{I}_{d-2})$ and $\boldsymbol{\Sigma}_Y = \text{diag}(3\boldsymbol{I}_2 + 3\theta\boldsymbol{B}_2, \boldsymbol{I}_{d-2})$, where $\boldsymbol{I}_d$ and $\boldsymbol{B}_d$ are identity and backward identity matrices with size $d \times d$, respectively. We generate different samples with respect to each $\theta$ and calculate the distance between them under different metrics and normalize these distances to ensure comparability. Fig. 4(a) presents the average distance across ten replicates. While most of the distances considered here demonstrate the curse-of-dimensionality through either parameter insensitivity or irregular fluctua-

tions, SEINT and ISEINT maintain smooth, consistent distance variations with respect to $\theta$, indicating superior dimensional stability.

**Computational Efficiency.** To further evaluate the efficiency of SEINT, we evaluate the runtime of all metrics across varying sample sizes $n$. We retain the data-generation setting from the high-dimensional experiment, but fix the dimension at $d = 4$ and set $\theta = 0.5$. For each $n$, we perform 10 independent runs and report the averaged results with error bars.

The computational efficiency analysis, presented in Fig. 4(b), reveals two key findings: (1) While SEINT and ISEINT show modest overhead for small $n$, they achieve superior scalability as sample sizes grow, outperforming alternatives by increasing margins; (2) The observed $n \log(n)$ complexity confirms the complexity analysis in Section 2.4 and algorithm 2.

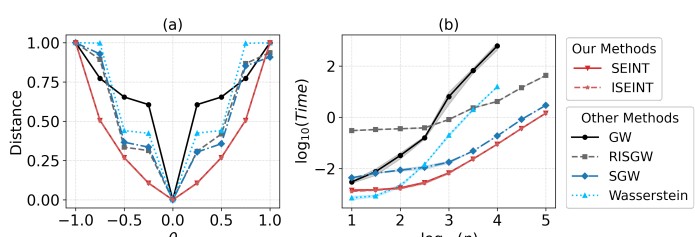

Figure 4: (a) Distances versus different $\theta$. (b) CPU time (s) versus different $n$ when $d = 4$.

## 3.2 AS A REGULARIZATION TERM

**3D Molecular Generation.** Recently, molecule generation (Zhou et al., 2023; Jin et al., 2020; Shi et al., 2021; Nichol & Dhariwal, 2021; Yue et al., 2025) has emerged as a prominent application of diffusion-based generative models (Wu et al., 2022; Huang et al., 2024; Ho et al., 2020). Among these, the E(3) Equivariant Diffusion Model (EDM) (Hoogeboom et al., 2022) is the first diffusion model to directly generate molecules in 3D space, showing strong performance across datasets. We also include UniGEM (Feng et al., 2024), an equivariant, geometry-aware model that jointly generates atom types and 3D coordinates, and serves as a competitive baseline for 3D molecular generation.

We integrate SEINT as a regularizer into the training objectives of both EDM and UniGEM, augmenting the standard $L_2$ loss (for coordinate denoising) with SEINT-based alignment terms; their relative weights are balanced via simple scale alignment. Following EDM and UniGEM, we also conduct experiments using the QM9 dataset (Ramakrishnan et al., 2014), a widely recognized benchmark containing approximately 130,000 small organic molecules, each composed of up to 9 heavy atoms. The dataset partitions follow Anderson et al. (2019), consisting of 100K training, 18K validation, and 13K test molecules. We further include GEOM-Drugs, the "drugs" partition of the GEOM dataset (Axelrod & Gomez-Bombarelli, 2022), which provides large-scale, energy-annotated 3D conformational ensembles for drug-like molecules. The detailed experimental settings are provided in the Appendix D.2.

Table 3: Pre-training results: Comparison of methods on atom stability (Atom.), molecular stability (Mol.), validity (Val.), and validity×uniqueness (V*U.), each drawing 10000 samples from the model.

| Backbones | EDM | | | | UniGEM | | | |
|---|---|---|---|---|---|---|---|---|
| Metrics | Atom.(%) | Mol.(%) | Val.(%) | V*U.(%) | Atom.(%) | Mol.(%) | Val.(%) | V*U.(%) |
| Data | 99.0 | 95.2 | 97.7 | 97.7 | 99.0 | 95.2 | 97.7 | 97.7 |
| Baseline | $98.7_{\pm0.1}$ | $87.2_{\pm0.4}$ | $93.4_{\pm0.4}$ | $91.7_{\pm0.3}$ | $98.9_{\pm0.1}$ | $89.4_{\pm0.1}$ | $94.6_{\pm0.1}$ | $92.8_{\pm0.1}$ |
| $L_2$-Norm$_{0.1}$ | $98.8_{\pm0.1}$ | $87.6_{\pm0.2}$ | $94.5_{\pm0.2}$ | $92.8_{\pm0.2}$ | $98.9_{\pm0.1}$ | $89.0_{\pm0.2}$ | $95.0_{\pm0.2}$ | $93.1_{\pm0.1}$ |
| $L_2$-Norm$_{0.2}$ | $98.7_{\pm0.1}$ | $87.2_{\pm0.1}$ | $93.7_{\pm0.3}$ | $92.1_{\pm0.3}$ | $98.7_{\pm0.1}$ | $85.3_{\pm0.4}$ | $93.9_{\pm0.5}$ | $92.2_{\pm0.4}$ |
| $L_2$-Norm$_{0.3}$ | $98.7_{\pm0.1}$ | $86.0_{\pm0.7}$ | $93.8_{\pm0.4}$ | $92.4_{\pm0.6}$ | $99.0_{\pm0.1}$ | $89.5_{\pm0.7}$ | $95.2_{\pm0.2}$ | $\mathbf{93.3_{\pm0.2}}$ |
| **SEINT$_{0.1}$** | $98.9_{\pm0.1}$ | $89.0_{\pm0.1}$ | $95.3_{\pm0.2}$ | $\mathbf{93.0_{\pm0.3}}$ | $99.1_{\pm0.1}$ | $91.0_{\pm0.3}$ | $95.5_{\pm0.3}$ | $92.8_{\pm0.3}$ |
| **SEINT$_{0.2}$** | $99.0_{\pm0.1}$ | $90.1_{\pm0.3}$ | $95.8_{\pm0.2}$ | $92.6_{\pm0.3}$ | $99.2_{\pm0.1}$ | $92.1_{\pm0.2}$ | $96.5_{\pm0.2}$ | $92.4_{\pm0.3}$ |
| **SEINT$_{0.3}$** | $\mathbf{99.1_{\pm0.1}}$ | $\mathbf{91.5_{\pm0.2}}$ | $\mathbf{96.5_{\pm0.2}}$ | $92.0_{\pm0.3}$ | $\mathbf{99.3_{\pm0.1}}$ | $\mathbf{93.5_{\pm0.2}}$ | $\mathbf{96.9_{\pm0.2}}$ | $91.8_{\pm0.3}$ |

[1] The notation SEINT$_\alpha$ indicates that the objective is $\mathcal{L} = \alpha\mathcal{L}_{\text{SEINT}} + (1 - \alpha)\mathcal{L}_{\text{MSE}}$.

[2] $L_2$-Norm: loss computed only on the coordinates.

Tables 3 and 4 report the performance of SEINT when applied as a regularization term in both pre-training and fine-tuning experiments. The results demonstrate that SEINT significantly enhances

chemical validity, atomic stability, and molecular stability. In particular, when the SEINT proportion is set to 0.3 in UniGEM, the model achieves state-of-the-art (SOTA) performance in both atomic stability and molecular stability. Although a slight reduction in uniqueness is observed, this trade-off is justified by the substantial improvements in validity, which is arguably the most critical factor for generating chemically plausible molecules.

Table 4: Fine-tuning results on QM9 and GEOM-Drugs.

| Datasets | QM9 | | | | GEOM-Drugs | | | |
|---|---|---|---|---|---|---|---|---|
| Metrics | Atom.(%) | Mol.(%) | Val.(%) | V*U.(%) | Atom.(%) | Mol.(%) | Val.(%) | V*U.(%) |
| UniGEM | 98.9 | 89.4 | 94.6 | 92.8 | 84.8 | 1.2 | 98.3 | 98.3 |
| $L_2$-Norm | 99.1 | 91.2 | 95.7 | _93.5_ | 84.4 | 1.3 | **99.2** | **99.2** |
| **SEINT$_{0.1}$** | _99.2_ | _92.4_ | _96.6_ | **93.5** | _86.7_ | _4.18_ | _98.5_ | _98.5_ |
| **SEINT$_{0.3}$** | **99.4** | **94.9** | **97.8** | 88.6 | **90.7** | **13.0** | 97.7 | 97.4 |

## 4 CONCLUSION

In this paper, we have introduced two SE($p$)-invariant representations: the Polar Transport Discrepancy (PTD) and the Distance-convoluted Polar Transport Discrepancy (DcPTD). We have proven that optimizing DcPTD via a 1D Optimal Transport formulation yields a strict metric on the space of isometry classes. Through a series of experiments, we have demonstrated the effectiveness of our proposed metric and its computational efficiency. Furthermore, it can be effectively employed as a regularization term in various complex shape learning tasks. Promising future directions include: (1) extending the loss function's regularization applications to generative modeling, computer vision, and multi-modal domains; and (2) designing novel learning architectures that capitalize on its distinctive characteristics.

## ACKNOWLEDGMENTS AND DISCLOSURE OF FUNDING

This work was supported by National Natural Science Foundation of China under Grant 92270110 and Grant 12471244, and National Key R&D Plan of China under Grant 2024YFA1016200.

## ETHICS STATEMENT

The datasets used in this paper are all publicly available and do not involve any ethical issues.

## REPRODUCIBILITY STATEMENT

We are committed to ensuring the reproducibility of our research. The source code for all experiments is provided in the supplementary materials, which can be used to reproduce the results presented in this paper. For the molecule generation experiment, we have included our key evaluation checkpoint. This checkpoint can also be reproduced using the provided training code. Furthermore, we have explicitly stated all key assumptions for the theorems presented. The sources or preprocessing scripts for all datasets used in this work are also provided.

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

# A  ADDITIONAL BACKGROUND

**Wasserstein Distance.**    Let $(\Omega, d)$ be a Polish metric space, and let $\mu_X, \mu_Y$ be two Borel probability measures on $(\Omega, \mathcal{B}(\Omega))$. The $p$-*Wasserstein distance* ($p \geq 1$) between $\mu_X$ and $\mu_Y$ is defined via the optimal transport problem:

$$\mathcal{W}_p(\mu_X, \mu_Y) := \left( \inf_{\pi \in \Pi(\mu_X, \mu_Y)} \int_{\Omega \times \Omega} d(x, y)^p \, d\pi(x, y) \right)^{\frac{1}{p}}. \tag{11}$$

When the space is the real line $(\mathbb{R}, |\cdot|)$, let $\mu_X, \mu_Y$ be probability measures on $(\mathbb{R}, \mathcal{B}(\mathbb{R}))$. Let $F_X(x) = \mu_X((-\infty, x])$ and $F_Y(y) = \mu_Y((-\infty, y])$ be their respective cumulative distribution functions (CDFs), and let $F_X^{-1}$ and $F_Y^{-1}$ be the corresponding quantile functions. The $p$-Wasserstein distance admits the closed-form expression:

$$\mathcal{W}_p(\mu_X, \mu_Y) = \left( \int_0^1 |F_X^{-1}(u) - F_Y^{-1}(u)|^p \, du \right)^{\frac{1}{p}}. \tag{12}$$

**Metric Measure Space.**    Following the work of (Mémoli, 2011; Sturm, 2023), we define a *metric measure (mm) space* via the triplet $(X, d_X, \mathcal{B}(X))$, where $(X, d_X)$ is a Polish (complete and separable) metric space and let $\mathcal{B}(X)$ be its Borel $\sigma$-algebra. Let $\mu_X$ be a probability measure on the measurable space $(X, \mathcal{B}(X))$. The *support* of $\mu_X$, denoted $\text{supp}(\mu_X)$, is the smallest closed set $C \subseteq X$ such that $\mu_X(C) = 1$.

**Gromov-Wasserstein Distance.**    Consider two metric measure spaces $(X, d_X, \mu_X)$ and $(Y, d_Y, \mu_Y)$. The Gromov-Wasserstein ($\mathcal{GW}$) distance compares their intrinsic metric-measure structures. For $p \geq 1$, the $p$-*Gromov-Wasserstein distance* is defined as:

$$\mathcal{GW}_p(\mu_X, \mu_Y) := \left( \inf_{\pi \in \Pi(\mu_X, \mu_Y)} \int_{X \times Y} \int_{X \times Y} |d_X(x, x') - d_Y(y, y')|^p \, d\pi(x, y) \, d\pi(x', y') \right)^{\frac{1}{p}}, \tag{13}$$

where the pairs $(x, y)$ and $(x', y')$ are drawn independently from the coupling measure $\pi$. The term $|d_X(x, x') - d_Y(y, y')|^p$ measures the distortion of pairwise distances under the coupling $\pi$.

**Computational Costs.**    When $\mu_X$ and $\mu_Y$ are empirical distributions supported on $m$ and $n$ samples respectively, computing the Wasserstein distance (11) corresponds to solving a linear program with $mn$ variables and $m + n$ constraints, incurring a time complexity of at least $\mathcal{O}((m + n)^3)$ (van den Brand et al., 2020). Applying entropy regularization (Cuturi, 2013) transforms the problem, enabling the Sinkhorn algorithm with a significantly lower per-iteration complexity of $\mathcal{O}(mn)$. Further acceleration techniques include those presented in (Altschuler et al., 2019; Li et al., 2023a), among others. In the specific case of the one-dimensional Wasserstein distance (12), computation is much faster via sorting, requiring only $\mathcal{O}(m \log m + n \log n)$ time. This efficiency in one dimension has motivated several fast OT algorithms that leverage projections onto 1D subspaces (Bonneel et al., 2015; Li et al., 2024), achieving considerable success. Computing the Gromov-Wasserstein distance (13) involves solving a non-convex quadratic program, potentially requiring up to $\mathcal{O}(m^2 n^2)$ time complexity for exact solutions. While recent fast algorithms (Li et al., 2023b; Scetbon et al., 2022; Ouyang et al., 2026) offer significant computational speedups, this acceleration often comes at the cost of preserving the metric structure. The overall computation for the Gromov-Wasserstein distance remains significantly more demanding compared to the Wasserstein distance.

**Discussion on convexity.**    Extrinsic SE($p$)-invariant approaches such as Softassign Procrustes Matching (SPM) (Rangarajan et al., 1997), Rotational Invariant Sliced Gromov-Wasserstein (RISGW) (Vayer et al., 2019), and Wasserstein Procrustes (Grave et al., 2019; Alvarez-Melis et al., 2019) fundamentally rely on the joint optimization of transport couplings and orthogonal transformations. This formulation leads to an inherently non-convex problem. Recent advances such as the Normalized Radon CDT (Beckmann et al., 2025) reduce multi-dimensional data into sets of one-dimensional projections through the Radon transform, which completely avoids the non-convex orthogonal alignment step.

For intrinsic methods, representative metrics such as Gromov–Hausdorff (GH) distance (Gromov, 1981) and Gromov–Wasserstein (GW) distance (Mémoli, 2011) achieve strong empirical performance

in applications such as shape matching. However, both metrics require solving highly non-convex optimization problems, which results in substantial computational complexity. Several accelerations have been proposed, including SGW(Vayer et al., 2019), EGW(Peyré et al., 2016), and Low-Rank GW(Scetbon et al., 2022), although none of these approaches change the fundamental non-convex nature of the original formulation. Another line of work focuses on optimizing convex upper (Li et al., 2023a) or lower bounds of GW (Sato et al., 2020). These relaxations avoid non-convex optimization, at the cost of losing the metric property.

In contrast, SEINT achieves SE($p$)-invariance, the metric property and linear computational complexity, while retaining convexity in its discrete formulation according to Eq. 15. The continuous version requires further theoretical investigation, which we consider a direction for future research.

## B  PROOF OF THEOREM 1

Before presenting the proof, we first state the following lemma.

**Lemma 1 (Gluing Lemma)** *For given measure spaces $(X, \mu_X)$, $(Y, \mu_Y)$, and $(Z, \mu_Z)$, and given couplings $\pi_{X,Z} \in \Pi(\mu_X, \mu_Z)$ and $\pi_{Y,Z} \in \Pi(\mu_Y, \mu_Z)$, there exists a joint coupling $\pi_{X,Y,Z} \in \Pi(\mu_X, \mu_Y, \mu_Z)$ such that its marginals satisfy*

$$(\text{proj}_{(X,Z)})_{\#}\pi_{X,Y,Z} = \pi_{X,Z}, \quad (\text{proj}_{(Y,Z)})_{\#}\pi_{X,Y,Z} = \pi_{Y,Z},$$

*and its $(X, Y)$-marginal, $(\text{proj}_{(X,Y)})_{\#}\pi_{X,Y,Z}$, is a coupling in $\Pi(\mu_X, \mu_Y)$.*

**Proof.**  Given the disintegrations $d\pi_{X,Z}(x, z) = d\pi_{X|Z=z}(x)d\mu_Z(z)$ and $d\pi_{Y,Z}(y, z) = d\pi_{Y|Z=z}(y)d\mu_Z(z)$, we define the joint coupling $\pi_{X,Y,Z}$ via its disintegration: $d\pi_{X,Y,Z}(x, y, z) = d\pi_{X|Z=z}(x)d\pi_{Y|Z=z}(y)d\mu_Z(z)$. This $\pi_{X,Y,Z}$ satisfies the required marginal conditions by construction. $\qquad\square$

For distributions on the real line ($\mathbb{R}$), we obtain an important corollary:

**Corollary 3** *Consider three probability distributions $\mu_X, \mu_Y, \mu_Z$ on $\mathbb{R}$. Let $\pi_{X,Z}$ and $\pi_{Y,Z}$ be the optimal couplings for the $\mathcal{W}_1$ distance between $(\mu_X, \mu_Z)$ and $(\mu_Y, \mu_Z)$, respectively. Let $\pi_{X,Y,Z}$ be the joint distribution obtained via the Gluing Lemma construction from $\pi_{X,Z}$ and $\pi_{Y,Z}$. Then, its $(X, Y)$-marginal, $(\text{proj}_{(X,Y)})_{\#}\pi_{X,Y,Z}$, is an optimal coupling for the $\mathcal{W}_1$ distance between $\mu_X$ and $\mu_Y$.*

**Proof.**  Let $F_X, F_Y, F_Z$ denote their respective cumulative distribution functions (CDFs), and $F_X^{-1}, F_Y^{-1}, F_Z^{-1}$ the corresponding quantile functions. The optimal couplings $\pi_{X,Z}, \pi_{Y,Z}$ can be represented as pushforward measures:

$$\pi_{X,Z} = (F_X^{-1} \circ F_Z, Id)_{\#}\mu_Z$$
$$\pi_{Y,Z} = (F_Y^{-1} \circ F_Z, Id)_{\#}\mu_Z$$

Applying the Gluing Lemma construction (effectively using $Z$ as the common random variable whose quantile $F_Z(Z)$ drives the others), we obtain the joint distribution

$$\pi_{X,Y,Z} = (F_X^{-1} \circ F_Z, F_Y^{-1} \circ F_Z, Id)_{\#}\mu_Z.$$

The $(X, Y)$-marginal is $(\text{proj}_{(X,Y)})_{\#}\pi_{X,Y,Z} = (F_X^{-1} \circ F_Z, F_Y^{-1} \circ F_Z)_{\#}\mu_Z$. Let $U = F_Z(Z)$. If $F_Z$ is continuous, $U$ is uniform on $[0, 1]$, and the marginal corresponds to the law of $(F_X^{-1}(U), F_Y^{-1}(U))$, which is the optimal quantile coupling between $\mu_X$ and $\mu_Y$. Thus, $(\text{proj}_{(X,Y)})_{\#}\pi_{X,Y,Z}$ is an optimal coupling between $\mu_X$ and $\mu_Y$. $\qquad\square$

To prove that SEINT defines a well-defined metric, we will demonstrate that it satisfies the Identity of Indiscernibles, Symmetry, and the Triangle Inequality properties.

### B.1  PROOF OF IDENTITY OF INDISCERNIBLES

Drawing inspiration from the characterization of isometry in the context of Gromov-Wasserstein distances, such as (Sturm, 2023, Lemma 1.10), it suffices to demonstrate the equivalence of the following two conditions:

    (i) $\mathcal{L}_{\text{SEINT}}(X, Y, \mu_X, \mu_Y) = 0$.

    (ii) There exists a coupling $\pi \in \Pi(\mu_X, \mu_Y)$ such that $d_X(x_0, x_1) = d_Y(y_0, y_1)$ holds for $\pi \otimes \pi$-almost every pair of points $((x_0, y_0), (x_1, y_1))$.

**We first prove the implication (i) $\Rightarrow$ (ii).** By definition, the condition $\mathcal{L}_{\text{SEINT}}(X, Y, \mu_X, \mu_Y) = 0$ implies that there exists an optimal coupling $\pi \in \text{Cpl}(\mu_X, \mu_Y)$, for any reference measure $\mu_Z$ and the corresponding optimal couplings $\pi_{X,Z}^* \in \Pi_{X,Z}^*$ and $\pi_{Y,Z}^* \in \Pi_{Y,Z}^*$ satisfying condition in equation 6, the equality $\phi_{\pi_{X,Z}^*}(x) = \phi_{\pi_{Y,Z}^*}(y)$ holds for all $(x, y) \in \text{Supp}(\pi)$.

The proof will proceed in three parts.

**Part 1**  For a measured Banach space $(X, \|\cdot\|_X, \mu_X)$, let $p_X : X \to \mathbb{R}_{\geq 0}$ denote the norm function, $p_X(x) := \|x\|_X$. We denote the distribution of this norm (its pushforward measure) by $\mu_{p_X} := (p_X)_{\#}\mu_X$. We first aim to prove that if $\mathcal{L}_{\text{SEINT}}(X, Y, \mu_X, \mu_Y) = 0$, then these distributions of norms must be equal, i.e., $\mu_{p_X} = \mu_{p_Y}$.

**Proof.** Assume for contradiction that the distributions $\mu_{p_X}$ and $\mu_{p_Y}$ are not identical.

Let the reference measure be $\mu_Z = \mu_{p_X}$. Consequently, it follows that $\zeta_{\pi_{X,Z}^*} = 0$ almost everywhere (a.e.).

Regarding $\zeta_{\pi_{Y,Z}^*}$, we argue that there must exist some $\epsilon > 0$ such that the set $\{y \in \mathcal{Y} : \zeta_{\pi_{Y,Z}^*}(y) > \epsilon\}$ has positive $\mu_Y$-measure. Otherwise, if $\zeta_{\pi_{X,Z}^*}(y) = 0$ a.e., then we would have:

$$0 = \mathbb{E}(\zeta_{\pi_{Y,Z}^*}) = \int \int |p_Y(y) - z| \, d\pi_{Z|Y}^*(z) d\mu_Y(y)$$

$$= \int \int |p_Y(y) - z| \, d\pi_{Y,Z}^*(x, y)$$

$$= \int \int |p_Y(y) - p_X(x)| \, d\pi_{p_X,Y}^*(x, y)$$

$$\geq \mathcal{W}_1(\mu_{p_X}, \mu_{p_Y}),$$

which implies $\mathcal{W}_1(\mu_{p_X}, \mu_{p_Y}) = 0$, and thus $\mu_{p_X} = \mu_{p_Y}$ a.e.. This contradicts our assumption. Let $\mathcal{Y}_\epsilon = \{y \in \mathcal{Y} : \zeta_{\pi_{Y,Z}^*}(y) > \epsilon\}$. We know $\mu_Y(\mathcal{Y}_\epsilon) > 0$.

From $\zeta_{\pi_{X,Z}^*}(x) = 0$ a.e., we infer that $\phi_{\pi_{X,Z}^*}(x) = 0$ a.e.. By the premise $\mathcal{L}_{\text{SEINT}}(X, Y, \mu_X, \mu_Y) = 0$, there exists a coupling $\pi \in \text{Cpl}(\mu_X, \mu_Y)$ such that $\phi_{\pi_{Y,Z}^*}(y) = \phi_{\pi_{X,Z}^*}(x) = 0$ for all $(x, y) \in \text{Supp}(\pi)$. This implies that $\phi_{\pi_{Y,Z}^*}(y) = 0$ for $\mu_Y$-almost every $y$. Therefore, for $\mu_Y$-a.e. $y \in \mathcal{Y}$, we have:

$$\phi_{\pi_{Y,Z}^*}(y) = \int d_Y(y, y') \zeta_{\pi_{Y,Z}^*}(y') \, d\mu_Y(y') = 0.$$

Furthermore, from this, we have:

$$0 = \int d_Y(y, y') \zeta_{\pi_{Y,Z}^*}(y') \, d\mu_Y(y') \geq \int_{\mathcal{Y}_\epsilon} d_Y(y, y') \zeta_{\pi_{Y,Z}^*}(y') \, d\mu_Y(y') \geq \int_{\mathcal{Y}_\epsilon} \epsilon \, d_Y(y, y') \, d\mu_Y(y').$$

Since $\epsilon > 0$ and assuming $d_Y(y, y') \geq 0$, this yields that for $\mu_Y$-a.e. $y \in \mathcal{Y}$, the integral $\int_{\mathcal{Y}_\epsilon} d_Y(y, y') \, d\mu_Y(y') = 0$. If $d$ is a metric and $\mu_Y(Y_\epsilon) > 0$, this implies

$$d_Y(y, y') = 0 \quad \text{for } \mu_Y\text{-a.e. } y \in \mathcal{Y} \text{ and } \mu_Y\text{-a.e. } y' \in Y_\epsilon.$$

Consequently, for $\mu_Y$-a.e. $y_1, y_2 \in \mathcal{Y}$, we can find $y' \in Y_\epsilon$ (more accurately, from a subset of $Y_\epsilon$ with full measure within $Y_\epsilon$) such that $d_Y(y_1, y') = 0$ and $d_Y(y_2, y') = 0$. By the triangle inequality (if $d$ is a metric or pseudometric):

$$d_Y(y_1, y_2) \leq d_Y(y_1, y') + d_Y(y_2, y') = 0 + 0 = 0 \quad \text{a.e.}$$

This forces $d_Y(y_1, y_2) = 0$ for $\mu_Y$-a.e. $y_1, y_2$.

A similar argument (potentially by choosing $\mu_Z = \mu_{p_Y}$) leads to $d(x_1, x_2) = 0$ a.e. for all $x_1, x_2 \in \mathcal{X}$. This implies that both $\mu_X$ and $\mu_Y$ are supported on sets where all points have zero

distance from each other. Consequently, their push-forward measures under the radical map, $\mu_{p_X}$ and $\mu_{p_Y}$, must be Dirac measures (point masses), i.e., $\mu_{p_X} = \delta_{c_X}$ and $\mu_{p_Y} = \delta_{c_Y}$ for some constants $c_X, c_Y \geq 0$. However, if $\mu_{p_X}$ and $\mu_{p_Y}$ are Dirac measures, we can consider $\mu_{p_X} = \mu_{p_Y}$ in the sense of translation invariance.

This contradicts our initial assumption that $\mu_{p_X}$ and $\mu_{p_Y}$ are different. Therefore, the assumption must be false, and we conclude that $\mu_{p_X} = \mu_{p_Y}$. $\qquad\square$

**Part 2** Next, we show that the coupling $\pi$ must satisfy $\mathbb{E}_\pi[|p_X(x) - p_Y(y)|] = 0$.

**Proof.** In the previous part, we established that $\mu_{p_X} = \mu_{p_Y}$ a.e.. Consider two specific choices for the reference measure $Z$. First, let $Z_1 = p_X(x) + 1$, then $\zeta_{\pi^*_{X,Z_1}}(x) = 1, \zeta_{\pi^*_{Y,Z_1}}(y) = 1$. This choice is valid and satisfies condition in equation 6 for any pair of optimal couplings $(\pi^*_{X,Z_1}, \pi^*_{Y,Z_1})$.

Second, let $Z_2 = \begin{cases} p_X(x), & p_X(x) = 0 \\ p_X(x) + 1, & p_X(x) > 0 \end{cases}$, then $\zeta_{\pi^*_{X,Z_2}}(x) = \mathbf{1}\{p_X(x) > 0\}$ and $\zeta_{\pi^*_{Y,Z_2}}(y) = \mathbf{1}\{p_Y(y) > 0\}$. This choice also holds for any optimal couplings $\pi^*_{X,Z_2}, \pi^*_{Y,Z_2}$ satisfying equation 6. Since $X$ and $Y$ are assumed to be centered, the optimal coupling $\pi$ satisfying equation 6 must map the origin to the origin (i.e., $x = 0 \iff y = 0$ for $\pi$-a.e. $(x,y)$). Consequently, we obtain $\mathbb{E}_\pi[|\zeta_{\pi^*_{X,Z_1}}(x) - \zeta_{\pi^*_{Y,Z_1}}(y)|] = \mathbb{E}_\pi[|1-1|] = 0$ and $\mathbb{E}_\pi[|\zeta_{\pi^*_{X,Z_2}}(x) - \zeta_{\pi^*_{Y,Z_2}}(y)|] = \mathbb{E}_\pi[|\mathbf{1}\{p_X(x) > 0\} - \mathbf{1}\{p_Y(y) > 0\}|] = 0$.

Consequently, for any reference distribution $\mu_Z$ and corresponding optimal couplings $\pi^*_{Y,Z}, \pi^*_{X,Z}$ satisfying equation 6, from the condition $\phi_{\pi^*_{X,Z}}(x) = \phi_{\pi^*_{Y,Z}}(y)$ for all $(x,y) \in \mathrm{Supp}(\pi)$, we have the equality of the transport maps:

$$\int d_X(x,x')\zeta_{\pi^*_{X,Z}}(x')\,d\mu_X(x') = \int d_Y(y,y')\zeta_{\pi^*_{Y,Z}}(y')\,d\mu_Y(y'). \tag{14}$$

This equality can be rewritten by integrating over the coupling $\pi$. Since $\zeta_{\pi^*_{X,Z}}(x') = \zeta_{\pi^*_{Y,Z}}(y')$ for $(x',y') \in \mathrm{Supp}(\pi)$, equation 14 implies:

$$\int (d_X(x,x') - d_Y(y,y'))\,\zeta_{\pi^*_{X,Z}}(x')\,d\pi(x',y') = 0.$$

Substituting the potentials corresponding to $Z_1$ and $Z_2$ into this relation, we obtain the following two equations:

$$\int (d_X(x,x') - d_Y(y,y'))\,d\pi(x',y') = 0$$

$$\int (d_X(x,x')\mathbf{1}\{p_X(x') > 0\} - d_Y(y,y')\mathbf{1}\{p_Y(y') > 0\})\,d\pi(x',y') = 0$$

$$\Rightarrow \int (d_X(x,x')\mathbf{1}\{x' = 0\} - d_Y(y,y')\mathbf{1}\{y' = 0\})\,d\pi(x',y') = 0$$

$$\Rightarrow d_X(x,0) - d_Y(y,0) = 0$$

Since $d_X(x,0) = p_X(x)$ and $d_Y(y,0) = p_Y(y)$, this implies $p_X(x) = p_Y(y)$ for $\pi$-almost every $(x,y)$. Consequently, $\mathbb{E}_\pi[|p_X(x) - p_Y(y)|] = 0$. $\qquad\square$

**Part 3** Based on the above conclusion, we can readily prove that $d_X(x_0,x_1) = d_Y(y_0,y_1)$ for any pairs $(x_0,y_0), (x_1,y_1) \in \mathrm{Supp}(\pi)$.

**Proof.** For any given $(x_0,y_0) \in \mathrm{Supp}(\pi)$, we can define a reference measure $Z_3 = \begin{cases} p_Y(y), & p_Y(y) \leq p_Y(y_0) \text{ and } y \neq y_0 \\ p_Y(y) + 1, & p_Y(y) > p_Y(y_0) \text{ or } y = y_0 \end{cases}$. For the given Polar transport discrepancy $\zeta_{\pi^*_{X,Z_3}}(x) = \mathbf{1}\{p_X(x) > p_X(x_0) \text{ or } x = x_0\}$, we can set $\zeta_{\pi^*_{Y,Z_3}}(y) = \mathbf{1}\{p_Y(y) > p_Y(y_0) \text{ or } y = y_0\}$. As $\mathbb{E}_\pi[|p_X(x) - p_Y(y)|] = 0$, it can be readily shown that $\mathbb{E}_\pi[|\zeta_{\pi^*_{X,Z_3}}(x) - \zeta_{\pi^*_{Y,Z_3}}(y)|] = 0$, thus $\pi^*_{Y,Z_3}$ satisfies equation 6 and is corresponded to $\pi^*_{X,Z_3}$. Therefore, for any given $(x_1,y_1) \in \mathrm{Supp}(\pi)$, substituting $\zeta_{\pi^*_{X,Z_3}}$ into equation 14 yields:

$$\int (d_X(x_1,x') - d_Y(y_1,y'))\,\mathbf{1}\{p_X(x') > p_X(x_0) \text{ or } x' = x_0\}\,d\pi(x',y') = 0.$$

Similarly, we can define $Z_4 = \begin{cases} p_Y(y), & p_Y(y) \le p_Y(y_0) \\ p_Y(y) + 1, & p_Y(y) > p_Y(y_0) \end{cases}$ In this case, the Polar transport discrepancys $\zeta_{\pi^*_{X,Z_4}}(x) = \mathbf{1}\{p_X(x) > p_X(x_0)\}$ and $\zeta_{\pi^*_{Y,Z_4}}(y) = \mathbf{1}\{p_Y(y) > p_Y(y_0)\}$ also satisfy $\mathbb{E}_\pi[|\zeta_{\pi^*_{X,Z_4}}(x) - \zeta_{\pi^*_{Y,Z_4}}(y)|] = 0$ and equation 6. Substituting this $\zeta_{\pi^*_{X,Z_4}}$ into equation 14 gives:

$$\int (d_X(x_1, x') - d_Y(y_1, y')) \mathbf{1}\{p_X(x') > p_X(x_0)\} \, d\pi(x', y') = 0.$$

Subtracting the second resulting equation from the first, we obtain:

$$\int (d_X(x_1, x') - d_Y(y_1, y')) \mathbf{1}\{x' = x_0\} \, d\pi(x', y') = 0$$

$$\Rightarrow \int d_X(x_1, x')\mathbf{1}\{x' = x_0\} \, d\mu_X(x') = \int d_Y(y_1, y')\mathbf{1}\{y' = y_0\} \, d\mu_Y(y')$$

$$\Rightarrow d_X(x_1, x_0) = d_Y(y_1, y_0)$$

Since the previous derivation holds for any pair $(x_1, y_1) \in \text{Supp}(\pi)$ and any pair $(x_0, y_0) \in \text{Supp}(\pi)$, we conclude that $d_X(x_1, x_0) = d_Y(y_1, y_0)$ for all $(x_1, y_1), (x_0, y_0) \in \text{Supp}(\pi)$. $\qquad\square$

**We next prove the implication (ii) $\Rightarrow$ (i).** Assume (ii) holds, meaning $(X, d_X, \mu_X)$ and $(Y, d_Y, \mu_Y)$ are isometric. According to (Sturm, 2023, Lemma 1.10), there exists a measurable map $f : X \to Y$ such that $f_\# \mu_X = \mu_Y$ and $d_Y(f(x), f(x')) = d_X(x, x')$ for $\mu_X \otimes \mu_X$-almost every $(x, x')$. Assume without loss of generality that $X$ and $Y$ are centered, and then the isometry $f$ maps $0 \in X$ to $0 \in Y$. Consequently, the distributions of the norms $\mu_{p_X}$ and $\mu_{p_Y}$ are identical. Let $\pi$ be the coupling induced by the isometry $f$, i.e., $\pi = (Id, f)_\# \mu_X$. Since $y = f(x)$ for $(x, y) \in \text{Supp}(\pi)$ and $f$ is an isometry preserving the origin (implying $p_Y(y) = p_X(x)$), we have $\mathbb{E}_\pi[|p_X(x) - p_Y(y)|] = 0$.

Now, consider an arbitrary reference measure $\mu_Z \in \mathcal{P}(\mathbb{R})$ and an associated optimal coupling $\pi^*_{X,Z} \in \Pi^*_{X,Z}$. By applying Lemma 1, we can construct a joint distribution $\pi_{X,Y,Z}$ on $X \times Y \times Z$ whose marginals on $X \times Y$ and $X \times Z$ are $\pi$ and $\pi^*_{X,Z}$, respectively. Let $\pi_{Y,Z} = (\text{proj}_{Y,Z})_\# \pi_{X,Y,Z}$ denote its marginal distribution on $Y \times Z$.

Since $\pi$ is an optimal coupling between $\mu_{p_X}$ and $\mu_{p_Y}$, and $\pi^*_{X,Z}$ is an optimal coupling between $\mu_{p_X}$ and $\mu_Z$, it follows from Corollary 3 that the constructed marginal $\pi_{Y,Z}$ is also an optimal coupling between $\mu_{p_Y}$ and $\mu_Z$. That is, $\pi_{Y,Z} \in \Pi^*_{Y,Z}$.

Furthermore, for any $(x, y) \in \text{Supp}(\pi)$, we have $y = f(x)$. The construction via the gluing lemma ensures that the conditional distribution of $Z$ given $(X, Y) = (x, y)$ depends only on $x$. This implies that the conditional distributions satisfy $\pi_{Z|Y=y} = \pi_{Z|X=x}$. Since the Polar transport discrepancys $l$ (Definition 1) depend on both the point ($x$ or $y$) and the conditional distribution of $Z$, the conditions $p_X(x) = p_Y(y)$ and $\pi_{Z|Y=y} = \pi_{Z|X=x}$ ensure that $\zeta_{\pi^*_{X,Z}}(x) = \zeta_{\pi_{Y,Z}}(y)$. Consequently, $\mathbb{E}_\pi[|\zeta_{\pi^*_{X,Z}}(x) - \zeta_{\pi_{Y,Z}}(y)|] = 0$. Thus, the chosen optimal coupling $\pi_{Y,Z} \in \Pi^*_{Y,Z}$ together satisfy the constraint condition in equation 6.

Therefore, we have:

$$\phi_{\pi^*_{Y,Z}}(y) = \phi_{\pi^*_{Y,Z}}(f(x))$$

$$= \int d_Y(f(x), y')\zeta_{\pi^*_{Y,Z}}(y') \, d\mu_Y(y')$$

$$= \int d_Y(f(x), f(x'))\zeta_{\pi^*_{Y,Z}}(f(x')) \, d\mu_X(x')$$

$$= \int d_X(x, x')\zeta_{\pi^*_{X,Z}}(x') \, d\mu_X(x')$$

$$= \phi_{\pi^*_{X,Z}}(x).$$

Since $\phi_{\pi^*_{X,Z}}(x) = \phi_{\pi^*_{Y,Z}}(y)$ holds for all $(x, y) \in \text{Supp}(\pi)$, the term $\mathbb{E}_\pi\left[|\phi_{\pi^*_{X,Z}}(x) - \phi_{\pi^*_{Y,Z}}(y)|^p\right]$ is zero for any choice of reference measure $\mu_Z$ and its corresponding couplings $\pi^*_{X,Z}, \pi_{Y,Z}$. Then the infimum over $\pi$ in equation 7 is zero, which means $\mathcal{L}_{\text{SEINT}}(X, Y, \mu_X, \mu_Y) = 0$. This completes the proof of (ii) $\Rightarrow$ (i).

### B.2 PROOF OF SYMMETRY AND TRIANGLE INEQUALITY

The symmetry of $\mathcal{L}_{\text{SEINT}}$ follows directly from the symmetry of the Wasserstein distance involved in its definition.

We now prove the Triangle Inequality. Consider three measured Banach spaces $(X, \|\cdot\|_X, \mu_X)$, $(Y, \|\cdot\|_Y, \mu_Y)$ and $(Z, \|\cdot\|_Z, \mu_Z)$, we can find corresponding SEINT optimal couplings $\pi_{X,Z} \in \Pi(\mu_X, \mu_Z)$ and $\pi_{Y,Z} \in \Pi(\mu_Y, \mu_Z)$. According to the Lemma 1, we can find a joint coupling $\pi_{X,Y,Z} \in \Pi(\mu_X, \mu_Y, \mu_Z)$ such that

$$(\text{proj}_{(X,Z)})_\# \pi_{X,Y,Z} = \pi_{X,Z}, \quad (\text{proj}_{(Y,Z)})_\# \pi_{X,Y,Z} = \pi_{Y,Z}.$$

Let $\pi_{X,Y} := (\text{proj}_{(X,Y)})_\# \pi_{X,Y,Z}$. Then we have $\pi_{X,Y} \in \Pi(\mu_X, \mu_Y)$.

Furthermore, we can find a reference measure $\mu_R$ and corresponding couplings $\pi^*_{X,R}, \pi^*_{Y,R}$ that satisfy condition in equation 6 (relative to $\pi_{X,Y}$), such that

$$\left( \mathbb{E}_{\pi_{X,Y}}[|\phi_{\pi^*_{X,R}}(x) - \phi_{\pi^*_{Y,R}}(y)|^p] \right)^{\frac{1}{p}} \geq \mathcal{L}_{\text{SEINT}}(X, Y, \mu_X, \mu_Y).$$

For the given $\pi_{X,Z}, \mu_R$ and the corresponding coupling $\pi^*_{X,R}$, we can find a corresponding coupling $\pi^{*'}_{Z,R}$ satisfies condition in 6. And similarlily to $\pi_{X,Z}, \mu_R, \pi^*_{X,R}$. Therefore, we have:

$$
\begin{aligned}
\mathcal{L}_{\text{SEINT}}(X, Y) &\leq \left( \mathbb{E}_{\pi_{X,Y}} \left[ |\phi_{\pi^*_{X,R}}(x) - \phi_{\pi^*_{Y,R}}(y)|^p \right] \right)^{\frac{1}{p}} \\
&= \left( \mathbb{E}_{\pi_{X,Z,Y}} \left[ |\phi_{\pi^*_{X,R}}(x) - \phi_{\pi^*_{Y,R}}(y)|^p \right] \right)^{\frac{1}{p}} \\
&\leq \left( \mathbb{E}_{\pi_{X,Z,Y}} \left[ (|\phi_{\pi^*_{X,R}}(x) - \phi_{\pi^*_{Z,R}}(z)| + |\phi_{\pi^*_{Z,R}}(z) - \phi_{\pi^*_{Y,R}}(y)|)^p \right] \right)^{\frac{1}{p}} \\
&\leq \left( \mathbb{E}_{\pi_{X,Z,Y}} \left[ |\phi_{\pi^*_{X,R}}(x) - \phi_{\pi^*_{Z,R}}(z)|^p \right] \right)^{\frac{1}{p}} + \left( \mathbb{E}_{\pi_{X,Z,Y}} \left[ |\phi_{\pi^*_{Z,R}}(z) - \phi_{\pi^*_{Y,R}}(y)|^p \right] \right)^{\frac{1}{p}} \\
&= \left( \mathbb{E}_{\pi_{X,Z}} \left[ |\phi_{\pi^*_{X,R}}(x) - \phi_{\pi^*_{Z,R}}(z)|^p \right] \right)^{\frac{1}{p}} + \left( \mathbb{E}_{\pi_{Z,Y}} \left[ |\phi_{\pi^*_{Z,R}}(z) - \phi_{\pi^*_{Y,R}}(y)|^p \right] \right)^{\frac{1}{p}} \\
&\leq \mathcal{L}_{\text{SEINT}}(X, Z) + \mathcal{L}_{\text{SEINT}}(Z, Y)
\end{aligned}
$$

### B.3 EXTENSIONS OF THE THEOREM

It is evident from the preceding proofs that our theorem can be extended to more general settings.

**Extension to Metric Spaces.** Specifically, the proofs primarily rely on the properties that the norm is non-negative and is zero only at the origin ($x = 0$). This suggests that the norm function $\|\cdot\|_X$ could potentially be replaced by any function $p_X : X \to \mathbb{R}_{\geq 0}$ that is positive definite (i.e., $p_X(x) \geq 0$ with $p_X(x) = 0$ if and only if $x = 0$, assuming 0 is uniquely defined). For instance, consider a metric measure space $(X, d_X, \mu_X)$. If we designate a base point $0 \in X$, we can define the function $p_X : X \to \mathbb{R}_{\geq 0}$ as the distance to this base point, $p_X(x) := d_X(x, 0)$. In this setting, it becomes apparent that if the metric measure spaces under comparison also satisfy the condition that relevant isometries are origin-preserving (mapping the base point 0 to the base point 0), then our proposed method remains a valid metric (or divergence) that respects these isometry classes within the context of metric measure spaces.

While extending from normed spaces to general metric measure spaces might seem straightforward without imposing further conditions, in reality, there is a subtle difference. For normed vector spaces, according to (Narici & Beckenstein, 2010, Theorem 9.1.2), if a surjective isometry between real normed spaces that maps the origin to the origin, such an isometry also maps the mean of a distribution to the mean of the transformed distribution. Furthermore, the preservation of the mean by origin-preserving isometries between normed spaces provides a crucial simplification: it guarantees that centering the distributions is sufficient for achieving a meaningful alignment. Consequently, we do not need a separate procedure to select the base point 0 beyond this standard centering process. However, for general metric measure spaces, which lack inherent vector space structure, the choice of the base point 0 becomes more critical and might require an explicit selection.

**Extension of the Distance Function in Normed Spaces.** The distance function used in our proofs is naturally the one induced by the norm of the underlying Banach space. However, we are not restricted to using this specific metric. We can replace the original distance $d_X(x_0, x_1) = \|x_0 - x_1\|_X$ with a powered version:

$$d_X^p(x_0, x_1) = \|x_0 - x_1\|_X^p, \quad \forall p > 0.$$

We examine how this change affects the key steps of our previous proofs:

In the proof establishing $\mu_{p_X} = \mu_{p_Y}$ (previously referred to as Part 1), the step involving the integral over $Y_\epsilon$ becomes $\int_{Y_\epsilon} \epsilon \, d_Y^p(y, y') \, d\mu_Y(y') = 0$. Since $d_Y^p(y, y') = \|y - y'\|_Y^p \geq 0$ and $\epsilon > 0$, this still implies $d_Y(y, y') = 0$ for $\mu_Y$-a.e. $y \in \mathcal{Y}$ and $\mu_Y$-a.e. In the proofs of the previous Part 2, the derivation relied on showing $d_X(x, 0) = d_Y(y, 0)$. If we instead obtain $d_X^p(x, 0) = d_Y^p(y, 0)$ (i.e., $\|x\|_X^p = \|y\|_Y^p$), this directly implies $d_X(x, 0) = d_Y(y, 0)$ since $p > 0$ and norms are non-negative. Similarly, in the proofs of Part 3, the derivation relied on showing $d_X(x_1, x_0) = d_Y(y_1, y_0)$. Obtaining $d_X^p(x_1, x_0) = d_Y^p(y_1, y_0)$ directly implies $d_X(x_1, x_0) = d_Y(y_1, y_0)$ for the same reason.

The proof of the triangle inequality for the SEINT distance itself operates on the extracted features $\phi_{\pi_{X,Z}^*}$ and the $L_p$ structure of the SEINT definition (e.g., equation 7), not directly on the properties of the function $d_X^p$. Therefore, this part of the argument is unaffected by whether $d_X^p$ itself satisfies the triangle inequality.

Consequently, regardless of the chosen value $p > 0$, even when $p > 1$ (for which $d_X^p(x_0, x_1)$ is generally not a metric due to violating the triangle inequality), the resulting SEINT formulation constructed using $d_X^p$ to define the DcPTD $t$ still satisfies the metric properties stated in Theorem 1.

### B.4 Proof of Corollary 2

Consider the discrete setting with inputs $\mathbf{X} = (\boldsymbol{x}_1, \ldots, \boldsymbol{x}_n)^\top \in \mathbb{R}^{n \times p}$ and $\mathbf{Y} = (\boldsymbol{y}_1, \ldots, \boldsymbol{y}_m)^\top \in \mathbb{R}^{m \times q}$, with corresponding probability vectors $\boldsymbol{p}_\mathbf{X}$ and $\boldsymbol{p}_\mathbf{Y}$. For simplicity, we can assume $m = n$ and uniform weights, as any discrete measure can be resampled to fit this structure. We further assume that the norms of the points within each distribution are distinct, i.e., $\|\boldsymbol{x}_i\| \neq \|\boldsymbol{x}_j\|$ for $i \neq j$. This allows us to define a unique ordering of the norms: $\|\boldsymbol{x}_{(1)}\| < \|\boldsymbol{x}_{(2)}\| < \cdots < \|\boldsymbol{x}_{(n)}\|$ and $\|\boldsymbol{y}_{(1)}\| < \|\boldsymbol{y}_{(2)}\| < \cdots < \|\boldsymbol{y}_{(n)}\|$. Under this assumption, the set of optimal couplings for the 1D norm distributions in equation 1 becomes a singleton, simplifying the analysis.

As the norms of $\mathbf{X}$ and $\mathbf{Y}$ are bounded by $M$, there exists $\epsilon > 0$, s.t. $\|\boldsymbol{x}_i\|, \|\boldsymbol{y}_j\| + \epsilon \leq M$. Following the construction of the reference measures $\mu_Z$ detailed in Appendix B.1, we can define a specific, minimal set of reference measures, which is denoted as $\tilde{\mathcal{P}}(\mathbb{R})$. This set can be constructed from a matrix $\tilde{\mathcal{P}}_X(\mathbb{R})$, where each column defines the support of a reference measure with uniform weights:

$$\tilde{\mathcal{P}}_X(\mathbb{R}) = \epsilon \begin{pmatrix} 1 & 0 & 0 & \cdots & 0 & 0 & 0 \\ 1 & 1 & 0 & \cdots & 0 & 0 & 0 \\ 1 & 1 & 1 & \cdots & 0 & 0 & 0 \\ \cdots & & & \cdots & & & \cdots \\ 1 & 1 & 1 & \cdots & 1 & 0 & 0 \\ 1 & 1 & 1 & \cdots & 1 & 1 & 0 \end{pmatrix} + \begin{pmatrix} \|\boldsymbol{x}_{(1)}\| \\ \|\boldsymbol{x}_{(2)}\| \\ \cdots \\ \|\boldsymbol{x}_{(n)}\| \end{pmatrix} \mathbf{1}_n^\top.$$

Similarly, we can construct $\tilde{\mathcal{P}}_Y(\mathbb{R})$. Let $\tilde{\mathcal{P}}(\mathbb{R})$ be the stacked matrix $(\tilde{\mathcal{P}}_X \quad \tilde{\mathcal{P}}_Y)$. As established in Appendix B.1, the following quantity defines a metric on this discrete space:

$$\tilde{\mathcal{L}}_{\text{SEINT}}(X, Y, \mu_X, \mu_Y) := \inf_{\pi \in \Pi(\mu_X, \mu_Y)} \sup_{\mu_Z \in \tilde{\mathcal{P}}(\mathbb{R})} \left( \mathbb{E}_\pi \left[ |\phi_{\pi_{X,Z}^*}(x) - \phi_{\pi_{Y,Z}^*}(y)|^p \right] \right)^{\frac{1}{p}}.$$

Now, consider the larger space of reference measures $\hat{\mathcal{P}}(\mathbb{R}) = \{\mu_\mathbf{Z} = \sum_{j=1}^k \frac{1}{k} \delta_{z_j} : \{z_j\}_{1 \leq j \leq k} \in [0, M]^k\}$ defined in the theorem. The corresponding dissimilarity is:

$$\hat{\mathcal{L}}_{\text{SEINT}}(X, Y, \mu_X, \mu_Y) := \inf_{\pi \in \Pi(\mu_X, \mu_Y)} \sup_{\mu_Z \in \hat{\mathcal{P}}(\mathbb{R})} \left( \mathbb{E}_\pi \left[ |\phi_{\pi_{X,Z}^*}(x) - \phi_{\pi_{Y,Z}^*}(y)|^p \right] \right)^{\frac{1}{p}}.$$

**Proof of corollary 2.** Given the condition that $\|\boldsymbol{x}_i\| + 1 \leq M$ and $\|\boldsymbol{y}_j\| + 1 \leq M$ for all $i, j$, it is clear that our deterministically constructed set of measures is a subset of the larger space, i.e.,

$\tilde{\mathcal{P}}(\mathbb{R}) \subseteq \hat{\mathcal{P}}(\mathbb{R})$. This implies that the supremum over the larger set must be greater than or equal to the supremum over the smaller set, leading to the inequality $\hat{\mathcal{L}}_{\text{SEINT}} \geq \tilde{\mathcal{L}}_{\text{SEINT}}$.

We can now prove the identity of indiscernibles for $\hat{\mathcal{L}}_{\text{SEINT}}$. If $\hat{\mathcal{L}}_{\text{SEINT}}(\mathbf{X}, \mathbf{Y}, \mu_{\mathbf{X}}, \mu_{\mathbf{Y}}) = 0$, then from the inequality, it must be that $\tilde{\mathcal{L}}_{\text{SEINT}}(\mathbf{X}, \mathbf{Y}, \mu_{\mathbf{X}}, \mu_{\mathbf{Y}}) = 0$. Since $\tilde{\mathcal{L}}_{\text{SEINT}}$ is a valid metric, this implies that $(\mathbf{X}, \|\cdot\|_{\mathbf{X}}, \mu_{\mathbf{X}})$ and $(\mathbf{Y}, \|\cdot\|_{\mathbf{Y}}, \mu_{\mathbf{Y}})$ are isometric. The reverse direction, along with symmetry and the triangle inequality, follows by the same logic as in the proof of Theorem 1. Therefore, we have shown that $\hat{\mathcal{L}}_{\text{SEINT}}$ is a well-defined metric for comparing discrete distributions in this space. $\square$

## C  IMPLEMENTATION DETAILS

In this section, we primarily discuss some implementation details of Algorithm 1, associated acceleration techniques, and the implementation of the expected algorithm.

### C.1  DETAILS OF ALGORITHM 1

**Whether to Standardize.**  In the proof of Theorem 1, we discussed that standardizing in a normed space avoids the need for explicit base point selection. Therefore, our standardization procedure first involves centering the data. Additionally, if the two distributions reside in the same ambient space, no further adjustments are necessary. Otherwise, we need to perform further scaling adjustments on both distributions, for example, by dividing by their respective standard deviations or a pooled standard deviation.

**Selection of $\mu_{\mathbf{Z}}$.**  Following the result of Corollary 2, the support points for the reference measure $\mu_Z$ are drawn from the space $[0, M]^k$. Specifically, we construct $\mu_{\mathbf{Z}}$ as a discrete uniform measure supported on $k$ points sampled i.i.d. from $\text{Unif}(0, M)$. In the special case where $m = n$ and the measures $\mu_{\mathbf{X}}$ and $\mu_{\mathbf{Y}}$ are discrete and uniform, the optimal coupling between their sorted norm distributions and the sorted support of $\mu_{\mathbf{Z}}$ can be obtained directly by matching sorted indices. In the more general case, the DcPTD can be efficiently computed using the North-West Corner Rule in conjunction with the formulation in equation 9. Regarding the choice of the upper bound $M$, our theoretical analysis in Corollary 2 establishes a requirement on the norms $\|\boldsymbol{x}\|$ for $\boldsymbol{x} \in \text{supp}(\mu_{\mathbf{X}})$. This suggests that $M$ should be related to the maximum observed norm (e.g., $M \approx \max \|\boldsymbol{x}\|$). However, to mitigate the potential impact of outliers that could unduly inflate this value, we typically set $M$ based on the statistics of the observed norms. For instance, a robust choice is: $M = \text{mean}(\|\boldsymbol{x}\|) + \text{std}(\|\boldsymbol{x}\|)$.

**Additional Scaling of $l$.**  Since our reference measure $\mu_Z$ is generated randomly, when the input point clouds themselves exhibit significant scale differences, the specific values sampled for $\mu_Z$ can potentially influence the resulting loss magnitude. For example, if a sample $z$ drawn for $\mu_Z$ has a value larger than all norms $\|\boldsymbol{y}\|$ for $\boldsymbol{y} \in \text{supp}(\mu_{\mathbf{Y}})$, then the larger this $z$ is, the larger the corresponding Polar transport discrepancy values $\zeta_z^{\mathbf{Y}}(\boldsymbol{y})$ (and consequently the DcPTD $\phi_z^{\mathbf{Y}}(\boldsymbol{y})$) will become. This inflates the overall loss value. However, this scaling effect might not significantly alter the choice of the optimal coupling $\mathbf{P}$. Therefore, to handle such potential scaling sensitivities related to extreme values in $\mu_Z$, we consider applying an additional standardization step to the Polar transport discrepancy vectors $\zeta_{\mathbf{Z}}^{\mathbf{X}}$ and $\zeta_{\mathbf{Z}}^{\mathbf{Y}}$.

It is worth noting that in the discrete setting, our loss can be expressed as follows:

$$
\begin{aligned}
\mathcal{L}_{\text{SEINT}} &= \min_{\mathbf{P} \in \Pi(\mu_{\mathbf{X}}, \mu_{\mathbf{Y}})} \max_{\mu_Z} \sum_{i=1}^{n} \sum_{j=1}^{m} \left\| \mathbf{C}_i^{\mathbf{X}} \zeta_{\mathbf{Z}}^{\mathbf{X}} - \mathbf{C}_j^{\mathbf{Y}} \zeta_{\mathbf{Z}}^{\mathbf{Y}} \right\|_2^2 \mathbf{P}_{ij} \\
&= \min_{\mathbf{P} \in \Pi(\mu_{\mathbf{X}}, \mu_{\mathbf{Y}})} \sum_{i=1}^{n} \sum_{j=1}^{m} \left( \mathbf{C}_i^{\mathbf{X}} \zeta_{\mathbf{Z_0}}^{\mathbf{X}} \zeta_{\mathbf{Z_0}}^{\mathbf{X}^\top} \mathbf{C}_j^{\mathbf{X}^\top} + \mathbf{C}_i^{\mathbf{Y}} \zeta_{\mathbf{Z_0}}^{\mathbf{Y}} \zeta_{\mathbf{Z_0}}^{\mathbf{Y}^\top} \mathbf{C}_j^{\mathbf{Y}^\top} - 2\mathbf{C}_i^{\mathbf{X}} \zeta_{\mathbf{Z_0}}^{\mathbf{X}} \zeta_{\mathbf{Z_0}}^{\mathbf{Y}^\top} \mathbf{C}_j^{\mathbf{Y}} \right) \\
&= \max_{\mathbf{P} \in \Pi(\mu_{\mathbf{X}}, \mu_{\mathbf{Y}})} \sum_{i=1}^{n} \sum_{j=1}^{m} 2\mathbf{C}_i^{\mathbf{X}} \zeta_{\mathbf{Z_0}}^{\mathbf{XY}} \mathbf{C}_j^{\mathbf{Y}^\top} \mathbf{P}_{ij} + \text{Const} \quad \left( \zeta_{\mathbf{Z_0}}^{\mathbf{XY}} = \zeta_{\mathbf{Z_0}}^{\mathbf{X}} \zeta_{\mathbf{Z_0}}^{\mathbf{Y}^\top} \right) \\
&= \max_{\mathbf{P} \in \Pi(\mu_{\mathbf{X}}, \mu_{\mathbf{Y}})} \sum_{i=1}^{n} \sum_{j=1}^{m} \sum_{i'=1}^{n} \sum_{j'=1}^{m} \mathbf{C}_{ii'}^{\mathbf{X}} \mathbf{C}_{jj'}^{\mathbf{Y}} \mathbf{P}_{ij} (\zeta_{\mathbf{Z_0}}^{\mathbf{XY}})_{i'j'} + \text{Const} .
\end{aligned}
$$

$$(15)$$

Simultaneously, we note that the discrete Gromov-Wasserstein (GW) loss can be expressed as:

$$
\begin{aligned}
\mathcal{GW}_2 &= \min_{\mathbf{P} \in \Pi(\mu_{\mathbf{X}}, \mu_{\mathbf{Y}})} \sum_{i=1}^{n} \sum_{j=1}^{m} \sum_{i'=1}^{n} \sum_{j'=1}^{m} \left( \mathbf{C}_{ii'}^{x} - \mathbf{C}_{jj'}^{\mathbf{Y}} \right)^2 \mathbf{P}_{ij} \mathbf{P}_{i'j'} \\
&= \min_{\mathbf{P} \in \Pi(\mu_{\mathbf{X}}, \mu_{\mathbf{Y}})} \sum_{i=1}^{n} \sum_{j=1}^{m} \sum_{i'=1}^{n} \sum_{j'=1}^{m} \left( \mathbf{C}_{ii'}^{\mathbf{X}\,2} + \mathbf{C}_{jj'}^{\mathbf{Y}\,2} - 2\mathbf{C}_{ii'}^{\mathbf{X}} \mathbf{C}_{jj'}^{\mathbf{Y}} \right) \mathbf{P}_{ij} \mathbf{P}_{i'j'} \\
&= \max_{\mathbf{P} \in \Pi(\mu_{\mathbf{X}}, \mu_{\mathbf{Y}})} \sum_{i=1}^{n} \sum_{j=1}^{m} \sum_{i'=1}^{n} \sum_{j'=1}^{m} 2\mathbf{C}_{ii'}^{\mathbf{X}} \mathbf{C}_{jj'}^{\mathbf{Y}} \mathbf{P}_{ij} \mathbf{P}_{i'j'} + \text{Const} .
\end{aligned}
$$

It is readily apparent that the computational structures of these two objective functions share similarities. A primary difference lies in the term involving the coupling matrix $\mathbf{P}$: our loss (in the final derived form) involves a term linear in $\mathbf{P}$ weighted by elements derived from $\zeta_{\mathbf{Z_0}}^{\mathbf{XY}}$, while the GW loss involves a term quadratic in $\mathbf{P}$. Motivated by this structural comparison and the goal of normalizing the influence of $\mu_{\mathbf{Z}}$, we naturally consider scaling the matrix $\zeta_{\mathbf{Z_0}}^{\mathbf{XY}} = \zeta_{\mathbf{Z_0}}^{\mathbf{X}} \zeta_{\mathbf{Z_0}}^{\mathbf{Y}^\top}$ such that its elements sum to 1. This is conceptually equivalent to appropriately normalizing the individual vectors $\zeta_{\mathbf{Z}}^{\mathbf{X}}$ and $\zeta_{\mathbf{Z}}^{\mathbf{Y}}$ derived from each $\mathbf{Z}$.

## C.2 ACCELERATED COMPUTATION FOR ALGORITHM 1

In the preceding discussion, we mentioned that for specific distance functions, our algorithm can be accelerated. Let us assume our distance function is the squared Euclidean distance, $d(x, y) = \|x - y\|_2^2$. As established previously, using the squared distance does not alter the fundamental metric properties of the resulting loss. Furthermore, for a data matrix $\mathbf{X} \in \mathbb{R}^{n \times p}$ representing points in Euclidean space, the matrix of squared pairwise Euclidean distances $\mathbf{C}^{\mathbf{X}}$ can be expressed as:

$$
\mathbf{C}^{\mathbf{X}} = \left( \|\boldsymbol{x}_i - \boldsymbol{x}_j\|_2^2 \right)_{ij} = \left( \boldsymbol{x}_i^\top \boldsymbol{x}_i + \boldsymbol{x}_j^\top \boldsymbol{x}_j - 2\boldsymbol{x}_i^\top \boldsymbol{x}_j \right)_{ij} = \|\boldsymbol{x}\|_{\text{row}}^2 \mathbf{1}_n^\top + \mathbf{1}_n \|\boldsymbol{x}\|_{\text{row}}^{2\top} - 2\mathbf{X}\mathbf{X}^\top
$$

where $\|\boldsymbol{x}\|_{\text{row}}^2$ denotes the vector whose $i$-th component is the squared Euclidean norm of the $i$-th row $\boldsymbol{x}_i$. Based on this explicit decomposition, we can derive the accelerated version, presented as follows.

As shown in Algorithm 2, by leveraging the explicit decomposition of the squared Euclidean distance matrix, we bypass the computationally intensive step of explicitly forming the full $n \times n$ and $m \times m$ distance matrices. Provided the condition $n \gg p$ and $m \gg q$ holds, this reduces the time complexity bottleneck from $\mathcal{O}(n^2 + m^2)$ to complexity dominated mainly by sorting of $\mathcal{O}(n \log n + m \log m)$. This algorithmic approach can be extended to any distance function whose corresponding distance matrix admits an efficient decomposition or allows for fast matrix-vector products.

## D ADDITIONAL EXPERIMENTS

### D.1 EFFECT OF NUMBER OF REFERENCE DISTRIBUTIONS ON SEINT.

We perform a sensitivity analysis of SEINT and ISEINT for the number of reference distributions. To this end, we use the horse and flamingo datasets from Section D.4 and evaluate how the loss

---

**Algorithm 2** SEINT Computation (Accelerated)

---

**Require:** Two datasets $\mathbf{X} \in \mathbb{R}^{n \times p}$ and $\mathbf{Y} \in \mathbb{R}^{m \times q}$, probability vectors $\boldsymbol{p_X} \in \Delta^{n-1}, \boldsymbol{p_Y} \in \Delta^{m-1}$, number of reference distributions $a$, support size for reference distributions $k$

1: Initialize the data $\mathbf{X}$ and $\mathbf{Y}$.
2: Generate $a$ random reference distributions $\mu_{\mathbf{Z}_1}, \mu_{\mathbf{Z}_2}, \ldots, \mu_{\mathbf{Z}_a}$ with $k$ samples
3: **for** $i \leftarrow 1$ to $a$ **do**
4: $\quad$ Compute $\zeta_{\mathbf{Z}_i}^{\mathbf{X}}$ using $\mathbf{X}, \boldsymbol{p_X}, \mu_{\mathbf{Z}_i}$ via equation 9 $\qquad \triangleright \mathcal{O}(n \log n + k \log k)$
5: $\quad$ Compute $\zeta_{\mathbf{Z}_i}^{\mathbf{Y}}$ using $\mathbf{Y}, \boldsymbol{p_Y}, \mu_{\mathbf{Z}_i}$ via equation 9 $\qquad \triangleright \mathcal{O}(m \log m + k \log k)$
6: $\quad \phi_{\mathbf{Z}_i}^{\mathbf{X}} \leftarrow \|\mathbf{x}\|_{\text{row}}^2 (\mathbf{1}_n^\top \zeta_{\mathbf{Z}_i}^{\mathbf{X}}) + \mathbf{1}_n(\|\mathbf{x}\|_{\text{row}}^{2\top} \zeta_{\mathbf{Z}_i}^{\mathbf{X}}) - 2\mathbf{X}(\mathbf{X}^\top \zeta_{\mathbf{Z}_i}^{\mathbf{X}})$
7: $\quad \phi_{\mathbf{Z}_i}^{\mathbf{Y}} \leftarrow \|\mathbf{y}\|_{\text{row}}^2 (\mathbf{1}_m^\top \zeta_{\mathbf{Z}_i}^{\mathbf{Y}}) + \mathbf{1}_m(\|\mathbf{y}\|_{\text{row}}^{2\top} \zeta_{\mathbf{Z}_i}^{\mathbf{Y}}) - 2\mathbf{Y}(\mathbf{Y}^\top \zeta_{\mathbf{Z}_i}^{\mathbf{Y}})$ $\qquad \triangleright \mathcal{O}(np + mq)$
8: $\quad$ Calculate the 1D-$\mathcal{W}_2$ loss: $\mathcal{L}_i \leftarrow \mathcal{W}_2\left((\phi_{\mathbf{Z}_i}^{\mathbf{X}})_{\#}\mu_{\mathbf{X}}, (\phi_{\mathbf{Z}_i}^{\mathbf{Y}})_{\#}\mu_{\mathbf{Y}}\right)$ $\quad \triangleright \mathcal{O}(n \log n + m \log m)$
9: **end for**
10: Set the final loss $\mathcal{L}_{\text{SEINT}} \leftarrow \max_{1 \leq i \leq a} \mathcal{L}_i$
11: **return** $\mathcal{L}_{\text{SEINT}}$

---

varies under different distribution numbers. From the figure 5, we observe that the loss for both SEINT and ISEINT gradually stabilizes as the number of reference distributions increases. This trend suggests that, given finite reference distributions, it is feasible to capture the maximum distinction between two input distributions. This observation not only demonstrates that SEINT is a well-defined metric but also indicates that its performance under a finite number of reference distributions closely approximates that obtained with an infinite number of references. In addition, we evaluated SEINT with the constructed reference measure $\mu_Z$ (see Appendix B.1). The results indicate that, when the sample size is sufficiently large, SEINT with a uniform reference distribution gradually exceeds the former, thereby providing further support for Corollary 2.

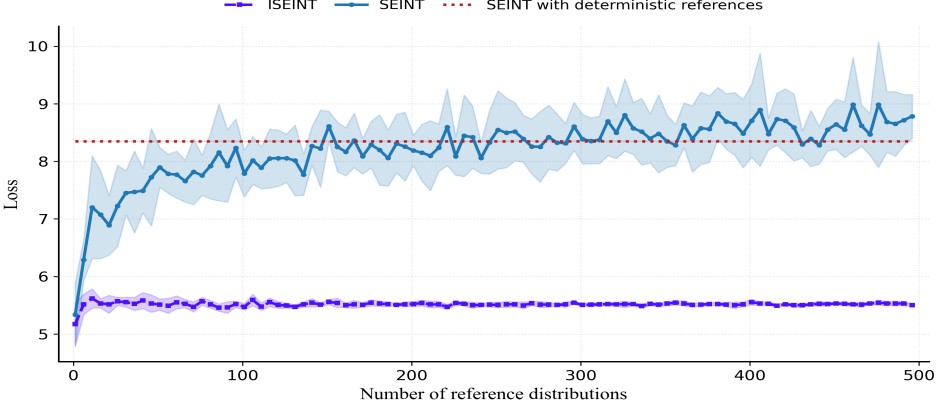

Figure 5: Effects of number of reference distributions on SEINT and ISEINT.

## D.2 MOLECULE GENERATION IN 3D

**Hyperparameter Configuration.** In the molecule generation experiments, we uniformly set the scale of the reference distributions to 3 and the number of reference distributions to 50. All other parameters were kept consistent with the settings in Hoogeboom et al. (2022). For the pretraining task on QM9, all methods were trained for 3000 epochs. For the fine-tuning experiments, our models were initialized from a baseline checkpoint. Specifically, for the QM9 dataset, we started from a checkpoint obtained after 2000 epochs of training and continued training for an additional 1000 epochs. For the GEOM-Drugs dataset, models were initialized from a checkpoint at 13 epochs and further trained for three additional epochs. The learning rate was set to $10^{-6}$ for QM9 and $10^{-5}$ for GEOM-Drugs.

**Additional results.** For the molecule generation task, we conducted additional experiments by adjusting the proportion of SEINT to 0.4 and 0.5. As shown in Table 5, both proportions significantly

outperform the baseline across validity, atom stability, and molecular stability metrics. However, we also observed a critical effect as the proportion of SEINT continued to increase. This limitation arises because SEINT utilizes only the first three-dimensional coordinate information, thus restricting the amount of information it can extract.

Table 5: Comparison of three regularization terms on validity, atom stability, molecular stability, and uniqueness, each drawing 10000 samples from the model.

| # Metrics | Validity (%) | Atom stable (%) | Mol stable (%) | Uniqueness (%) |
|---|---|---|---|---|
| EDM | 93.4±0.4 | 98.7 ±0.1 | 87.2 ±0.4 | 98.2±0.2 |
| SEINT$_{0.4 \text{ (ours)}}$ | 96.6±0.1 | 99.2±0.0 | 91.1±0.4 | 93.2±0.5 |
| SEINT$_{0.5 \text{ (ours)}}$ | 96.7±0.3 | **99.2±0.1** | **91.8 ±0.4** | 92.0±0.2 |

Furthermore, we observed that despite the improvements in validity, atom stability, and molecular stability, the uniqueness metric gradually decreased. We argue that there exists a trade-off between validity and uniqueness, with the two metrics exhibiting a degree of complementarity. Among them, validity is relatively more critical, as it ensures the chemical plausibility of the generated molecules. Therefore, we consider the decrease in uniqueness to be acceptable if it results from an improvement in validity.

### D.3 Masked Autoencoders for Point Cloud

The success of Masked Autoencoding (MAE) (He et al., 2022a; Xie et al., 2022; Baevski et al., 2022) in NLP and computer vision has inspired its extension to 3D data, with Point-MAE (Pang et al., 2022) emerging as the pioneering approach for point cloud reconstruction. Building on this, we augment Point-MAE by integrating SEINT and ISEINT as regularizers with the standard Chamfer Distance loss. Our evaluation protocol involves: (1) pretraining on ShapeNet (Chang et al., 2015), followed by (2) finetuning on ModelNet40 (1k), ModelNet40 (8k) (Wu et al., 2015), Scan-Objonly, and ScanObjectNN's most difficult configuration (PB-T50-RS) (Uy et al., 2019).

The results shown in Table 6 confirm that SEINT and ISEINT regularization consistently enhance classification accuracy over the baseline.

Table 6: Comparisons for various methods on their accuracy (%).

| Methods | PB-T50-RS | ModelNet40(1k) | Scan-Objonly | ModelNet40(8k) |
|---|---|---|---|---|
| Point-MAE | 84.67 | 93.07 | 87.60 | 93.52 |
| SEINT$_{0.1 \text{ (ours)}}$ | 84.87 | 93.52 | 88.47 | 93.68 |
| ISEINT$_{0.1 \text{ (ours)}}$ | **85.39** | **93.68** | **88.64** | **93.92** |

[1] The notation SEINT$_{0.1}$ indicates that the objective loss function $\mathcal{L} = 0.1\mathcal{L}_{\text{SEINT}} + 0.9\mathcal{L}_{\text{Point-MAE}}$.

**Hyperparameter Configuration.** In the point cloud reconstruction task, we consistently set the scale of reference distributions to 3 and the number of reference distributions to 100. All other parameters were kept the same as those used in Pang et al. (2022).

### D.4 Empirical Experiments of Continuity of Metric

Continuity is one of the fundamental properties of a metric. Although we have not yet established a formal proof that SEINT satisfies continuity, we can still empirically examine its behavior under gradually varying distributions. We selected three distinct 3D shapes from the mesh data of (Sumner & Popović, 2004): elephant, horse, and flamingo. Given a shape $X$, we generate progressively perturbed versions by linearly interpolating it with Gaussian noise $\varepsilon \sim \mathcal{N}(0, \sigma^2)$, with $\sigma = 0.2$. At each step $t$, the interpolated shape is defined as:

$$X_t = (1 - \alpha_t) \cdot X + \alpha_t \cdot \varepsilon, \quad \text{where } \alpha_t = 0.005t.$$

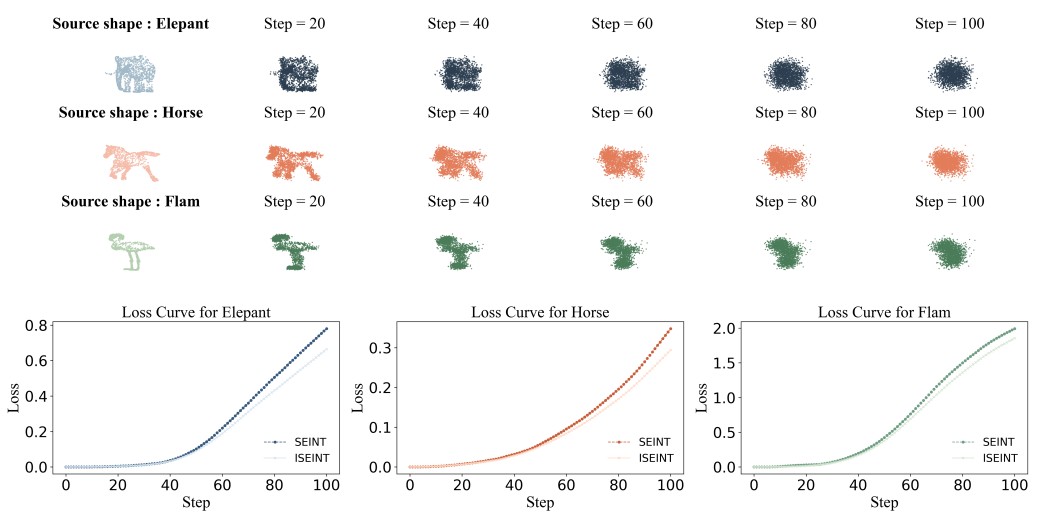

Figure 6: Loss curve across different levels of noise.

Figure 6 illustrates the SEINT and ISEINT loss values computed between the original and noisy point clouds throughout the noise accumulation process.

As shown in Figure 6, the loss between the source shapes and their corresponding noisy versions increases smoothly as the noise level grows, providing preliminary evidence that SEINT responds continuously to distribution changes.

### D.5 ADDITIONAL RESULTS FOR SE($p$)-INVARIANCE

To further validate the novelty and performance advantages of SEINT, we added additional baselines for comparison, including LRGW(Scetbon et al., 2022), PGW(Kerdoncuff et al., 2021), SFGW(Piening & Beinert, 2025), AE(Sato et al., 2020), and QGW(Chowdhury et al., 2021). We set the number of points in each point cloud to 1000, and the other experimental setup follows the same configuration as the "SE($p$)-Invariance" section in the main text. Table 7 reports the KNN classification accuracy under different neighborhood sizes, along with the corresponding runtime for each method.

Table 7: Classification accuracy (%) across different numbers of neighbors (k).

| Method | $\text{Acc}_{(k=1)}$ | $\text{Acc}_{(k=5)}$ | $\text{Acc}_{(k=10)}$ | Time (ms) | SE($p$)-Inv | Complexity |
|---|---|---|---|---|---|---|
| LRGW | $2.3_{\pm 1.94}$ | $2.1_{\pm 1.77}$ | $2.0_{\pm 2.00}$ | 1350.36 | ✓ | $\mathcal{O}(n^2 r)$ |
| PGW | $45.6_{\pm 1.36}$ | $42.9_{\pm 1.83}$ | $40.1_{\pm 1.20}$ | 693.47 | ✓ | $\mathcal{O}(n^2)$ |
| S(F)GW | $100.0_{\pm 0.00}$ | $100.0_{\pm 0.00}$ | $100.0_{\pm 0.00}$ | 485.19 | ✓ | $\mathcal{O}(n^2 \log n)$ |
| AE | $2.5_{\pm 0.89}$ | $2.5_{\pm 0.89}$ | $2.5_{\pm 0.89}$ | 406.25 | ✓ | $\mathcal{O}(n^2 \log n)$ |
| QGW | $5.0_{\pm 0.45}$ | $5.0_{\pm 0.45}$ | $5.0_{\pm 0.45}$ | 242.50 | ✓ | $\mathcal{O}(m^3 \log m)$ |
| **SEINT (Ours)** | $\mathbf{100.0_{\pm 0.00}}$ | $\mathbf{100.0_{\pm 0.00}}$ | $\mathbf{100.0_{\pm 0.00}}$ | **4.46** | ✓ | $\mathcal{O}(n \log n)$ |

[1] All reported times are CPU times and denote the **per-run runtime** of each method.
[2] $n$ denotes the sample size, and $m$ represents the number of quantization blocks used in QGW, and $r$ specifies the rank parameter in LRGW.
[3] The number of SEINT references and the rank of LRGW are both set to 50, whereas all other methods are run with their default parameters.

From Table 7, SEINT achieves the best overall performance in both accuracy and runtime. Among the additional baselines, only SFGW attains comparable accuracy. It is crucial to clarify that SEINT and SFGW employ fundamentally different representation mechanisms. Specifically, SEINT first utilizes polar length information to compute the optimal transport against a reference, yielding the PTDs; it subsequently convolves the PTD with the distance metric to derive the DcPTDs as

the final representation. In contrast, SFGW directly extracts information from the sorted distance matrix. This latter approach inevitably imposes a time complexity of $\mathcal{O}(n^2 \log n)$. Consequently, in practical scenarios, SEINT is significantly faster, demonstrating a distinct advantage in computational efficiency.

## D.6 Additional Results for Capability on High-Dimensional Data

To further evaluate the ability of SEINT to handle more complex high-dimensional data, we extend the previous Gaussian setting to a mixture-Gaussian framework. Specifically, we define

$$\mu = \mathcal{N}(0, I_d), \qquad \nu_\theta = 0.5\,\mathcal{N}(0, I_d) + 0.5\,\mathcal{N}\left(0,\ (1-\theta)I_d + \theta\,\frac{1_d 1_d^\top}{d}\right),$$

where $1_d$ denotes the all-ones vector and $\theta \in [0, 1]$. We generate samples over a range of $\theta$ values and examine how the estimated distances $D(\mu, \nu_\theta)$ vary with $\theta$, which enables a systematic comparison of how different methods respond as the target distribution gradually departs from the baseline distribution.

We consider two representative high-dimensional regimes, $n = 200, p = 50$ and $n = 50, p = 200$. Figure 7 presents the corresponding results.

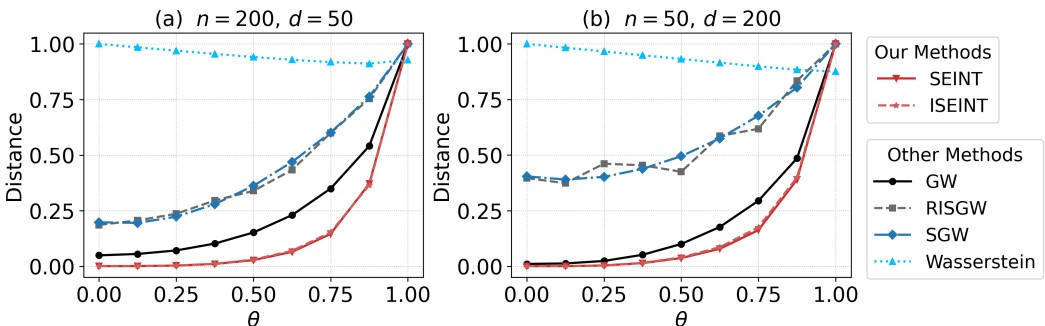

Figure 7: Behavior of SEINT under high-dimensional settings: $(n = 200, p = 50)$ and $(n = 50, p = 200)$.

As shown in Fig. 7, SEINT exhibits a consistent and well-behaved trend in both settings: the loss increases smoothly as the discrepancy between the distributions grows. This suggests that SEINT is capable of accurately capturing distributional differences even in challenging high-dimensional scenarios.

## D.7 Effect of Number of Reference Points on SEINT.

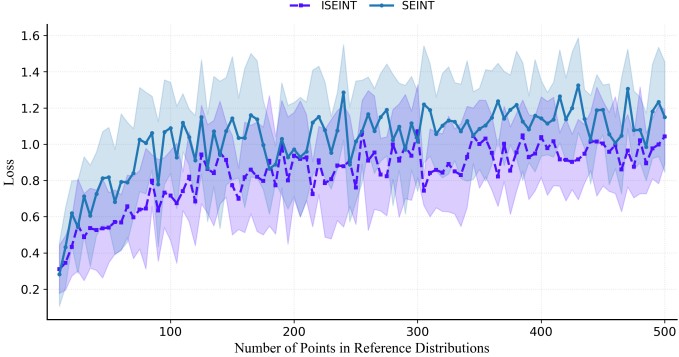

Figure 8: Effect of Number of Reference Points on SEINT.

To further examine how the number of reference points affects SEINT, we fix the sample size at $n = 2000$ and set the number of references to 100, while varying the number of points within each reference distribution. Fig. 8 reports the resulting loss values of SEINT under this setting.

From the Fig. 8, we observe that as the number of reference points increases, the SEINT loss gradually stabilizes. This indicates that SEINT requires only a relatively small number of reference points to achieve stable performance, demonstrating its robustness with respect to the choice of the scale of the distribution reference.

### D.8 MDS VISUALIZATION

To further validate the performance of SEINT, we included a visualization experiment (Vayer et al., 2019) based on the distance matrices of the horse running sequence. The experiment consists of two settings: one utilizing original (unrotated) point clouds and another where random rotations are applied.

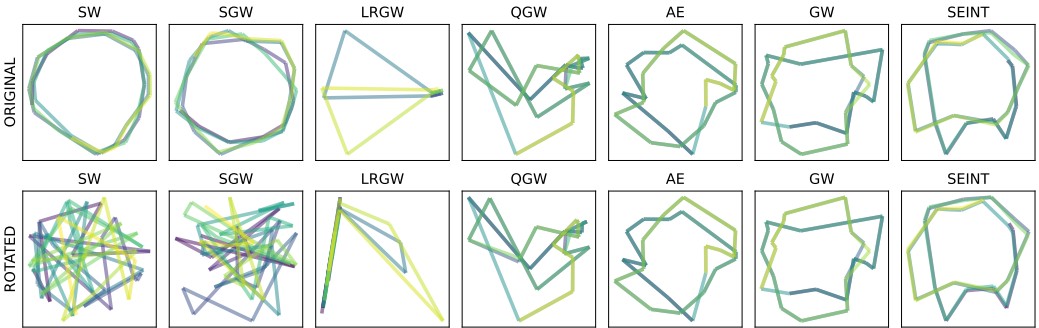

Figure 9: MDS result on horse gallop data.

As shown in the Fig. 9, for the unrotated data, SW, SGW, AE, GW, and SEINT all exhibit a regular, closed-loop pattern. However, after applying random rotations, only AE, GW, and SEINT preserve this original pattern. This result empirically demonstrates that SW and SGW lack rotation invariance, whereas SEINT preserves this crucial property.

### THE USE OF LARGE LANGUAGE MODELS

We acknowledge the use of Large Language Models in the preparation of this manuscript. Their role was strictly limited to functioning as a tool for language polishing and improving readability. We affirm that all definitions and theorems presented in this paper are the original intellectual contributions of the authors. The core code is intricately linked to the algorithm we propose, and the experiments were designed specifically to investigate the properties of our method. We take full responsibility for the integrity of this work, which is free from any form of academic misconduct.

