# OpenReview forum: "An Efficient SE(p)-Invariant Transport Metric Driven by Polar Transport Discrepancy-based Representation"
_ICLR.cc/2026/Conference — ICLR 2026 Poster_

### Official Review · Reviewer_dTTa · 2025-10-28

**Soundness:** 2
**Presentation:** 3
**Contribution:** 3
**Rating:** 4
**Confidence:** 3

**Summary:**

The authors propose a method to compare structured objects, e.g., graphs, shapes, etc. This is a hard task that is not solvable with standard optimal transport (OT) tools, but that requires tools such as the Gromov-Wasserstein (GW) distance (Memoli, 2019). Notably, the GW distance and its variants are computationally challenging. While more efficient methods exist, they often have severe shortcomings. As a remedy, the authors propose a method that relies on comparing two structure objects via pairwise distance matrices. In particular, they sample from a target reference 1D distribution to then map each distance matrix to 1D distribution. Then, they can compute the efficient 1d Wasserstein distance between the resulting 1D projected distance distributions. From that point of view, it can be considered as an interesting adaptation of sliced GW.

I think that this is a really interesting idea. While slicing is an important method in the field of OT, the so-called 'sliced GW' (SGW, Vayer et al., 2019)  method has shortcomings. While there are other very recent 'sliced GW' approaches out there that have been missed by the authors (Piening et al., 2025), the authors certainly showed the advantages over the SGW.

However, my main criticism is a lack of experimental validation. The experiments are very synthetic. As this is often the case with GW, this is not my main concern, however. More importantly, the authors failed to quantitatively compare their method against suitable alternatives. They mentioned GW (Memoli, 2011), entropic GW (Peyre et al., 2016), and low-rank GW (Scetbon, 2022), but did not really benchmark against them. The horse sequence is a nice visualization, but does not replace benchmarks on shape/graph classification.

Therefore, I feel as if this is a really nice idea, but the paper lacks substance beyond the introduction of the new metric. In particular, I would have expected either more theoretical contributions or a solid benchmarking to vote for acceptance.

**Strengths:**

- A good and interesting idea that not everyone would have come up with.
- A nice supplementary section on the OT background
- A interesting discussion of possible extensions in the supplement
- The code in the supplementary is accessible and mostly complete. That is very helpful!
- A nice introduction and categorization of existing methods.
- A well-motivated methodological section.
- Well-done visualizations, especially Fig. 1 & 2
- Proof section is well-structure. While the main theoretical contribution (Metric property) seems rather simple, the ‘identity of indiscernibles’ is not straightforward.

**Weaknesses:**

*Title*
- I have my doubts about the title. Despite background knowledge, the lengthy title was more confusing than helpful for my initial understanding. I would recommend dropping ‘DRIVEN BY POLAR TRANSPORT DISCREPANCY-BASED REPRESENTATION’ since ‘polar transport’ is not even well-known in the OT community, the word ‘transport’ appears in the title twice, and it would make the title more understandable overall. This is only a minor remark and question of style, however.

*Introduction*:

The introduction is well-written and relies on helpful categorization of existing methods. However, the authors did leave out a lot of modern methods.

- While the broad classification visualized in Figure 1 gives a nice overview and I would mostly agree with it. However, I feel as if the authors missed key references:
- - ‘Extrinsic Strategy’: While I acknowledge the authors’ efforts in citing key references from ‘90s, more recent research in the same direction has been overlooked, see papers on ‘Wasserstein Procrustes’ methods (Grave et al., 2019; Alvarez-Melis et al, 2019) and extensions beyond Euclidean spaces (Alaya et al, 2022; Beier et al., 2025). While these methods indeed rely on costly iterative, non-convex optimization iterations, another line of research (Beckmann et al., 2025) enables *efficient* comparison of Euclidean data via the ‘Normalized Radon CDT’.
 - - ‘Intrinsic Strategy’: Again, Memoli’s Gromov-Wasserstein (GW) paper from 2011 is a very important paper for comparing geometries. Nevertheless, a lot of effort has gone into efficient adaptations of this approach. While the paper mentions GW upper bounds (Li et al., 2023b; Peyré et al., 2016; Scetbon et al., 2022), GW lower bounds (Sato et al., 2020; Piening et al., 2025) have not been considered in the current submission, but they seem to enable rather efficient comparison of intrinsic geometries, while avoiding the non-convex optimization schemes of SGW/RISGW/GW/EGW/low-rank GW.

- Table 1: While this Table gives a nice overview, I have a few comments (from less important to more important):
- - 1.) The complexity of EMD solvers depends on the problem. I think that for $n \neq m$ the complexity is not directly cubic, but in $\mathcal{O}(n^3 log n)$.
- - 2.) The complexity of some metrics (especially sliced ones) is also dimension-dependent in a certain sense, whereas this is not the case for all of them. As an example, this should be the case of RISGW vs. GW.
- - 3.) A key disadvantage of RISGW, GW, and other methods is that they rely on non-convex solvers.
- - Conclusion: While the table is helpful for the reader, I feel as if further information should be given about convexity/convergence guarantees and dimensional dependence.

- Please note the ‘Sliced Gromov-Wasserstein’ paper has been authored by ‘Titouan Vayer’ and should probably be cited as ‘Vayer et al.’. To my knowledge, the incorrect citation as ‘Titouan et al’ is a persistent error on Google Scholar.

- I would like to point out that the aforementioned sliced GW variant (Piening et al, 2025) employs similar ideas based on projecting pairwise distance matrices to 1D distributions and computing 1D Wasserstein distances. While there are certainly differences, I think that they should be (briefly) discussed, maybe in the background section.

*Experiments*:
The experiments are doing a good job of illustrating key properties. However, there is most certainly a large comparative gap.
- The authors only compared against ‘standard’ OT metrics	 (which are not supposed to be isometry-invariant) and variations of sliced GW (SGW, Vayer et al, 2019). However: 1.) SGW is not isometry-invariant and 2.) the (RI)SGW algorithms has some theoretical problems (see the correction note on arXiv and (Beinert et al., 2022).
I feel as if there should be a benchmarking against another ‘extrinsic’ metric, e.g., ‘Wasserstein Procrustes’ (Grave et al., 2019), against GW solvers (Memoli; Peyre; Scetbon; Sato; Kerdoncuff; Chowdhury …), and against another isometry-invariant slicing-based metric (Piening et al., 2025; Beckmann et al., 2025).
- The presented experiments are highly synthetic. 1.) The horse-gallop experiment is a nice visualization, but not really suitable for benchmarking.  I would also recommend citing existing variations of this experiment, e.g., (Vayer et al., 2019). Moreover, the high-dimensional experiment is rather ‘easy’ as it only relies on normal distributions.
- The experiment on graph generative models is interesting, but again, it is only a very simple ‘benchmarking’ (SEINT vs L2 regularization)

**Questions:**

- Why are some entries in Table 1 red and others black?
- Why is (E)GW accuracy missing from Table 2? Almost all other reported distances are not isometry-invariant, and the one exception (RISGW) relies on solving a highly non-convex optimization problem.
- Section D.4 ‘Metric Consistency’: I do not know what the term ‘metric consistency’ is supposed to mean here. I feel as if the reported experiments illustrate the *continuity* of the distance. While it is good to see that the empirical estimator displays continuity, I would be more interested in a proof of the continuity. Did you look into it, or did I miss it in the paper?
- Sliced Wasserstein (SW) suffers from a curse of dependence. Can you comment on the dimensional dependence?
- Often the quality of Monte Carlo methods, e.g., SW, scales with $\frac{1}{\sqrt(\text{Projection Number}$ (Nadjahi et al., 2020). This is a simple result of Hölder’s inequality. Could you get similar results for your estimates?
- Can you incorporate graphs describing geodesic distances similar to experiments in (Memoli, 2011), (Beier et al., 2022), or (Piening et al, 2025)?
- Can you extend your setting to the setting of 'Fused Gromov--Wasserstein' (Vayer et al, 2020)?

---

> ### Author Response · Authors · 2025-11-21
> **Response to Reviewer dTTa (Part 1 / 3)**
>
> Thank you for your thoughtful feedback. Below, we provide responses to your comments in the following section and we have modified the manuscript to address these issues. (Changes are highlighted in blue in the updated PDF.)
>
> ---
> ### **Writing and Presentation**
> In the revised manuscript, we have refined the Introduction and addressed typos. We also cited and discussed the related work as recommended. Please find our detailed responses to your specific comments below:
>
> - **Q1: The issue of title.**
>
>     We originally adopted the title ''driven by polar transport discrepancy'' to highlight the technical characteristics of our method, distinguishing it from other ''SE(3)-invariant'' approaches. The repetition of ''transport'' was intentionally designed to reflect the SEINT workflow, which inherently involves two distinct optimal transport steps. However, we acknowledge that ''polar transport'' is a term introduced for the first time in this work, which might unintentionally affect the title's readability. We appreciate your feedback and will carefully consider revising it in the future version.
>
> - **Q2: Modern methods in Introduction.**
>
>     In the Introduction, we have added citations and discussions for the relevant literature regarding ''Extrinsic Strategy'' and ''Intrinsic Strategy.'' Additionally, we have provided further supplementary details in the Appendix A (Line 909-927).
>
> - **Q3: The complexity of EMD.**
>
>     We have now revised the computational complexity of EMD in the manuscript, correcting it from $O(n^3)$ to $O(n^3) \sim O(n^3 \log n)$.
>
> - **Q4: The dimensional complexity in Table 1.**
>
>     The time complexity of certain methods depends on the dimension $p$ indeed. **However, throughout our paper, we assume $n \gg p$.** Consequently, our analysis primarily focused on the complexity with respect to the sample size $n$. As noted in Algorithm 1, SEINT's complexity regarding the dimension is $O(n^2+np)$ (where $p$ mainly accounts for the computation of norms), and the accelerated method is $O(n\log n + np)$ (where $p$ mainly accounts for the computation of DcPTDs), which does not affect the overall complexity under the assumption that $n \gg p$. To avoid any ambiguity, we have added an explanatory note in Table 1 (Line 92) regarding the relationship between sample size and dimension.
>
> - **Q5: GW rely on non-convex solvers.**
>
>     We have included a discussion of the convexity of the various methods in Appendix A (Line 909-927).
>
> - **Q6: Citation of 'Titouan Vayer'**
>
>     Thank you for the correction. We have updated the author information for ''Sliced Gromov-Wasserstein'' in our references.
>
> - **Q7: Discussions on SFGW**
>
>     We have incorporated this reference into the Introduction and discussed the differences between SEINT and SFGW in the Introduction and Appendix D.5 (Lines 1507-1514). In particular, **SEINT and SFGW employ fundamentally different representation mechanisms.** Specifically, SEINT first utilizes polar length information to compute the optimal transport against a reference, yielding the PTDs; it subsequently convolves the PTD with the distance metric to derive the DcPTDs as the final representation. In contrast, SFGW directly extracts information from the sorted distance matrix, which leads to information loss and fails to satisfy metric properties. Furthermore, the latter approach inevitably imposes a time complexity of $O(n^2\log n)$, resulting in much longer runtime (Lines 1494-1505).
>
> - **Q8: Red entries in Table 1**
>
>     We apologize for the confusion. We originally marked the lowest complexity ($O(n \log n)$) and ''Yes'' entries in red to clearly distinguish the strengths of each method. However, we agree that this inconsistent coloring caused unnecessary confusion. In the revised version, we have standardized all these entries to black.
>
> ---
>
> ### **Theoretical Property**
> On the theoretical front, we proposed a novel metric derived from SE(p) invariance. Since SEINT is a newly proposed metric, a detailed investigation into its theoretical properties, e.g., its continuity and its dependence on dimension, is still pending. Our current work provides only a preliminary investigation of these properties through numerical experiments, which may address some of your initial concerns. We will conduct a rigorous analysis of these fundamental properties in our future work.
> Please find our detailed responses to your specific comments below:

---

> ### Author Response · Authors · 2025-11-21
> **Response to Reviewer dTTa (Part 2 / 3)**
>
> - **Q1: Monte Carlo results**
>
>     Regarding the curse of dimensionality, we have partially addressed this concern numerically in the section ''Capability on High-Dimensional Data''(Line 422-431). As illustrated in Figure 4(b), SEINT remains sensitive in distinguishing even subtle distributional differences in high dimensions, demonstrating its potential in high-dimensional settings. We have further extended our experiments to the Mixture Gaussians setting to simulate more complex scenarios. As detailed in the Appendix D.6 (Line 1515-1546), ''Supplementary Experiments on High-Dimensional Data,'' SEINT's performance curve exhibits a reasonable pattern in both the $n > p$ and $p > n$ regimes.
>
> - **Q2: Metric Consistency**
>
>     Our intention in using the term ''metric consistency'' was to illustrate that for two initially identical distributions, the SEINT loss curve shows a consistent, gradual increase as noise is progressively introduced to one of them. **In essence, this experiment is indeed an empirical verification of the continuity of SEINT as a metric.** To completely resolve the ambiguity, we have renamed this experiment in the revised manuscript to ''Empirical Experiments on the Continuity of the Metric.''
>
> ---
>
> ### **Experiment**
> We have significantly enriched our evaluation. First, we included additional baselines in the point cloud classification task to further verify the effectiveness of our method. Second, we added an MDS visualization of the distance matrix for the ''horse running sequence'' in the Appendix D.8 (Line 1571-1594). Third, we conducted classification experiments on the graphs describing the geodesic distances dataset to demonstrate SEINT's performance within the metric space. Finally, we extended the setting of our high-dimensional experiments to include Mixture Gaussians models, verifying SEINT's capability in handling more complex scenarios. Detailed responses to your specific comments are shown below:
>
> - **Q1: Add a benchmarking against other metrics.**
>
>     We have incorporated several additional baselines for comparison, including SFGW[1], PGW[2], AE[3], QGW[4], and LRGW[5]. In particular, we set the number of points in the point cloud to 1000, while all other settings remain consistent with the ''SE(p) Invariance'' section in the main text. The table below presents the classification accuracy (using KNN) and the single-run computation time for each method. SEINT achieves the highest overall performance. Although SFGW achieves the same accuracy, SEINT demonstrates a significant advantage in computation time.
>
>     |Method|Acc(k=1)|Acc(k=5)|Acc(k=10)|Time(ms)|
>     |-|-:|-:|-:|-:|
>     |SFGW|100.00±0.00|100.00±0.00|100.00±0.00|485.185|
>     |LrGW|2.30±1.94|2.10±1.77|2.00±2.00|1350.36|
>     |PGW|45.60±1.36|42.90±1.83|40.10±1.20|693.47|
>     |QGW|5.00±0.45|5.00±0.45|5.00±0.45|242.50|
>     |AE|2.50±0.89|2.50±0.89|2.50±0.89|15.82|
>     |**SEINT(Ours)**|100.00±0.00|100.00±0.00|100.00±0.00|4.46|
>
>
> - **Q2: More tasks on horse-gallop data**
>
>     We have included an MDS visualization experiment [6] on the distance matrices of the horse running sequence in Appendix D.8 (Linee 1571-1594). To facilitate efficient computation, we performed equidistant sampling to select approximately 1000 points for comparison.
>
>     The experiment consists of two settings: one with original (unrotated) point clouds and another with randomly rotated point clouds. As shown in the Fig.9, for the unrotated data, SW, SGW, AE, GW, and SEINT all exhibit a regular, closed-loop pattern. However, after applying random rotations, only AE, GW, and SEINT preserve this original pattern. This empirically demonstrates that SW and SGW lack rotation invariance, whereas SEINT robustly maintains it.
>
> - **Q3: Add high-dimensional experiments on non-normal distributions.**
>
>     We have further extended the settings of our high-dimensional experiments in Appendix D.6 (Line 1515-1546). Specifically, building upon the original Gaussian framework, we expanded the target distributions to Mixture Gaussians models. We defined the distributions as follows:
>     $$
>     \mu = \mathcal{N}(0, I_d), \qquad \nu_\theta = 0.5\,\mathcal{N}(0, I_d) + 0.5\,\mathcal{N}\bigl(0,\,(1-\theta) I_d + \theta\, 1_d 1_d^\top / d\bigr),
>     $$
>     where $1_d$ denotes the all-ones vector and $\theta \in [0,1]$. We generated samples across a range of $\theta$ values and examined the distance curves $D(\mu, \nu_\theta)$ produced by various methods to evaluate their ability to capture the progressive deviation from the baseline distribution.
>     We evaluated the performance under two distinct settings: $n=200, p=50$ ($n > p$) and $n=50, p=200$ ($n < p$). The results indicate that in both high-dimensional scenarios, SEINT exhibits a consistent pattern where the loss increases monotonically as the distributional discrepancy grows. This confirms SEINT's capability to accurately characterize distribution differences, even in high-dimensional settings.

---

> ### Author Response · Authors · 2025-11-21
> **Response to Reviewer dTTa (Part 3 / 3)**
>
> - **Q4: Graph generative models is a simple 'benchmarking'**
>
>     The motivation of this experiment is to verify whether SEINT contributes meaningful structural information as a regularizer. The results demonstrate that by incorporating SEINT into the existing baseline, we achieved **State-of-the-Art (SOTA) performance**, which validates the effectiveness of our method. In addition, **L2 regularization is not included here as a competitor, but rather as an ablation study**. Its purpose is to verify that our performance gains do not stem merely from increasing the proportion of coordinate information. Instead, the significant improvement over L2 proves that SEINT succeeds by capturing intrinsic geometric information that simple coordinate information cannot.
>
> - **Q5: (E)GW accuracy missing**
>
>     **The reason for the absence of GW and EGW results is computational feasibility**: The total time required to complete the full testing for these methods would exceed 2000 hours based on their per-iteration runtime, which is intractable. We have added an explanatory note below the table clarifying why these accuracy results are absent to avoid ambiguity. Following your suggestion, we have incorporated additional rotation-invariant baselines, including SFGW, PGW, AE, QGW, and LRGW. In the Table 7 (Line 1494) and revised version of Table 2 (Line 378), we have also added a column to explicitly indicate whether each method possesses SE(p) invariance.
>
> - **Q6: Experiments on Graphs describing geodesic distances**
>
>     While SEINT primarily focuses on structured data within Euclidean space, **it can be effectively generalized to metric spaces**. Specifically, by replacing the $L_2$ norm with the distance to the Barycenter, SEINT also works in metric spaces while preserving its metric properties. Although this exploration is beyond the main scope of our current work, to address your question, we have added 3-KNN classification experiments on the Animals and FAUST datasets, following the protocols in [1]. The table below shows that SEINT and ISEINT consistently achieve superior performance. Note that, unlike the Euclidean norm, the geodesic distance cannot be explicitly decomposed. Consequently, the complexity of SEINT in this setting becomes $O(n^2)$. Nevertheless, even with this constraint, SEINT maintains a distinct computational advantage over most comparative methods.
>
>     |distance|animals acc.(%)|animals time(ms)|FAUST-500 acc.(%)|FAUST-500 time(ms)|FAUST-1000 acc.(%)|FAUST-1000 time(ms)|
>     |-|-|-|-|-|-|-|
>     |S(F)GW|99.1±1.2|3.3|36.7±5.8|26.3|40.4±6.0|35.9|
>     |AE|97.9±1.8|1.6|37.8±5.6|13.9|41.6±6.0|15.9|
>     |GW|100.0±0.0|8.3|29.2±4.3|662.5|33.3±5.2|1915.0|
>     |QGW|57.7±5.1|4.8|13.1±3.5|32.6|13.3±3.3|47.7|
>     |PGW|86.6±3.6|133.4|19.1±3.9|695.9|18.6±3.8|1025.9|
>     |**SEINT**|99.3±1.3|0.8|39.3±5.7|5.9|42.3±6.4|19.2|
>     |**ISEINT**|99.4±1.4|0.6|39.4±5.8|4.9|42.9±6.7|13.7|
>
>
> - **Q7: Extend to Fused Version?**
>
>     **We can extend SEINT to the 'Fused' setting.** Specifically, following the approach in [1], we fuse the geometric structures (DcPTDs) and the features into a single representation, and then compute the Sliced Wasserstein distance between these resulting mixtures. We tested this fused algorithm (with $\alpha = 0.5$) on the Animals and FAUST classification tasks. The table below confirms that FSEINT also achieves consistently strong performance under the given mixing ratio, validating the effectiveness of our fused approach.
>
>     |distance|animals acc.(%)|animals time(ms)|FAUST-500 acc.(%)|FAUST-500 time(ms)|FAUST-1000 acc.(%)|FAUST-1000 time(ms)|
>     |-|-|-|-|-|-|-|
>     |SFGW|98.9±1.2|3.2|25.7±3.8|18.8|25.3±4.0|715.8|
>     |FAE|97.8±1.8|1.6|32.5±4.6|9.1|32.4±4.4|676.4|
>     |FGW|100.0±0.0|9.0|22.6±4.5|246.0|221.3±4.3|3227.7|
>     |**FSEINT**|99.5±1.0|2.7|42.5±5.8|15.2|43.4±5.8|42.0|
>
>     This extension was not included in the main text because SEINT is originally designed to yield a distance metric that is strictly rotation-invariant. The 'fused' setting sacrifices this crucial property. Therefore, we plan to delve deeper into this extension in future work, where we can rigorously analyze the resulting trade-offs.
>
>
> In summary, we hope the above response resolves your concerns and helps you re-evaluate our work. Feel free to contact us if you have any other questions.
>
> **Reference**
>
> [1] A novel sliced fused Gromov-Wasserstein distance. 2025
>
> [2] Sampled Gromov Gasserstein. 2021
>
> [3] Fast and robust comparison of probability measures in heterogeneous spaces. 2020
>
> [4] Quantized Gromov-Gasserstein. 2021
>
> [5] Linear-time Gromov Gasserstein distances using low rank couplings and costs. 2022
>
> [6] Sliced Gromov-Gasserstein. 2019

---

> > ### Comment · Reviewer_dTTa · 2025-11-22
> > **Initial Acknowledgment of Replies**
> >
> > I thank the authors for putting so much time and effort into their rebuttal. While I may take a couple of days to go through the updated document and the very detailed answers, I am very optimistic that I will raise my score eventually. Based on my current understanding, this seems to be good work! :)
> >
> > Regarding the new experiments, I re-read [1] and noted that in their experiments their S(F)GW and AE have similar runtimes and performance. In the Table from your Q6 answer, this impression is confirmed, but in the Table of Q1 there is a stark difference (wrt to performance and runtime). Could you help me understand this phenomenon?
> >
> > Further, I would like to emphasize that performance and runtime of many GW solvers relies heavily on certain parameters, e.g., entropic regularization for Sinkhorn, number of slicing projections for S(F)GW, quantization degree in quantized GW. As this hyperparameter choices often balances runtime and performance, I think that it would be good to add a somewhat prominent remark about the importance of parameter choices.
> >
> > While this is certainly out of scope, I also want to emphasize that I would be very interested in follow-up work on the geometry, including barycenter computations.

---

> > > ### Author Response · Authors · 2025-11-22
> > > **Thanks**
> > >
> > > Thank you for the prompt response! We are delighted about your positive feedback. We are happy to address your additional concerns as follows:
> > >
> > > 1. Regarding the runtimes of S(F)GW and AE: To expedite the experimental process for Q1, we initially utilized an optimized, faster implementation of AE (which yields identical accuracy results). This led to the discrepancy with the runtime reported in Q6. We have since reverted to the  AE implementation found in the S(F)GW code repository and re-evaluated the runtime. The updated measurement is 406.246 ms per run, which aligns consistently with the results in Q6. We apologize for any confusion this inconsistency may have caused.
> > >
> > > 2. Regarding hyperparameter choices for other methods: Given the extensive number of baselines compared, we adopted the default hyperparameters provided in their respective official implementations for most methods. We acknowledge that this setting might not yield the absolute peak performance for every individual baseline; however, performing a comprehensive hyperparameter search for all competing methods falls outside the scope of this work. We will include a clarification note regarding this in our future version. Nevertheless, we believe these experiments sufficiently demonstrate the effectiveness of SEINT as a SE(p)-invariant metric.
> > >
> > > We once again thank you for recognizing the value of our work!

---

> > > > ### Comment · Reviewer_dTTa · 2025-11-24
> > > > **Update**
> > > >
> > > > My concerns about the experiment were mainly addressed, and I raised my score.

---

> > > > > ### Author Response · Authors · 2025-11-24
> > > > > **Thanks to the Reviewer**
> > > > >
> > > > > We sincerely thank the reviewer for taking the time to re-evaluate our manuscript and update the score.

---

> > > ### Author Response · Authors · 2025-12-01
> > > **PDF Update**
> > >
> > > Dear Reviewer dTTa,
> > >
> > > We have updated the PDF manuscript and included a clarification note regarding the hyperparameter selection for the comparative methods in Table 7 (Lines 1493–1507). We appreciate your valuable suggestion.

---

### Official Review · Reviewer_SoMY · 2025-10-30

**Soundness:** 4
**Presentation:** 4
**Contribution:** 4
**Rating:** 8
**Confidence:** 3

**Summary:**

The paper proposes a computationally efficient approach for a representation based metric  for the problem of comparing probability distributions that lie in different Banach spaces invariant to orthogonal transformations. The method involves looking at the optimal transport between one-dimensional measures over norms and each distributions  norm distribution. The contribution to the transport cost for each point in the distribution provides a non-negative value, and this is distance convolved to provide a representation that is isometric consistent and dimension independent. Solving an optimal transport between the two measures using this representation provides a way to align the two distributions.

**Strengths:**

The paper addresses a challenging problem that can be applied to point cloud registration and outperforms Gromov-Wasserstein distance. It is clearly written, original, and results are strong.

**Weaknesses:**

There are a few minor points to clarify.

**Questions:**

In the algorithm the choice of generating reference distributions as uniform (line 290) is not discussed. Corollary 2 does not mention how the supports $\{z_j\}_{j=1}^k$ are chosen. Is uniform the simplest, why not quantiles of of the empirical norms?

Line 52 "it may sacrifice the metric properties of the distance"  but in Table 1 it is stated that it isn't a metric. Thus 'may' is not correct.

Line 147 should state $\lVert X \rVert_X$ and $\mu_Z$

Line 157, the notation with norm and $\cdot$ breaking across the line is hard to parse.

Line 278, it was not previously assumed that the space is $\mathbb{R}^{n\times p}$. While the special Euclidean invariant assumes this, at other times it would seem that the theory is applicable to complex-valued vector space (or possibly other field) which is not necessary in the discrete setting. This should be clarified.

Minor extra space around the superscript for the footnote 1.

---

> ### Author Response · Authors · 2025-11-21
> **Response to Reviewer SoMY**
>
> We appreciate the reviewer's positive assessment regarding the clarity, originality, and strong empirical results of our work. Please find our responses to your questions below, along with corresponding edits in the revised manuscript. (Changes are highlighted in blue in the updated PDF.)
>
> - **On Q1: The Reference Selection**
>
>     In the proof of our theorem, reliance on specific reference measures is necessary. Consequently, we expanded the selection space for the reference measures based on these requirements. By taking the supremum over this expanded space, we obtain a ''least favorable measure,'' ensuring that the theoretically required measure remains a valid candidate. To practically encompass these necessary measures, we adopted a simple hypercube constraint, which we found to be sufficiently effective in our experiments. Conversely, if we were to consider the quantiles of the empirical norms as the selection space, it might fail to include the specific reference measure required for Theorem 1.
>
> - **On Q2-Q4 \& Q6: Typos**
>
>     We appreciate your detailed observations and corrections. We have corrected these typos in the revised manuscript.
>
> - **On Q5: Scope of the Banach Space**
>
>     Although SEINT is applicable to more general measured Banach spaces (e.g., ensuring unitary invariance in complex-valued vector spaces), our applications in this paper primarily focus on metrics for real-valued data. We will explicitly clarify this distinction at the beginning of Section 2.4.
>
> In summary, we hope the above response resolves your concerns and solidifies your support for our paper in the following discussion phases. Thanks again for your consideration.

---

> > ### Comment · Reviewer_SoMY · 2025-11-26
> >
> > Thank you for the response to my and other reviewers concerns. It is very solid work and maintaining my score.

---

> > > ### Author Response · Authors · 2025-11-26
> > > **Thank you**
> > >
> > > We sincerely thank you for your recognition of our work.

---

### Official Review · Reviewer_4D2S · 2025-10-31

**Soundness:** 3
**Presentation:** 3
**Contribution:** 3
**Rating:** 4
**Confidence:** 4

**Summary:**

This paper proposes a new variant of an Optimal Transport (OT) metric that is Special Euclidean group-Invariant (SE(p)). The main idea of the authors is to proceed in two steps. The first one is to align the original raw samples with a certain reference measure, where the cost function is defined as a difference between a given vector and a coordinate of the vector from the reference measure in 1D. The second step is to convolve the obtained embedding with the raw cost matrix and to use the obtained embedding to define a cost function for the resulting 1D OT problem. This allows the metric to be SE(p), yet it is to be computed efficiently in practice. The algorithmic implementation then takes the sup over the reference measures to calculate the final value of the metric. The experiments, both qualitative and quantitative, showcase the desirable properties of the proposed metric.

**Strengths:**

1.	The proposed metric seems to be novel
2.	Proposed metric has metric properties and is cheap to compute
3.	Experimental evaluations provide a multi-faceted view on the proposed metric.

**Weaknesses:**

1.	The introduction of PTD and SEINT is very hard to parse in the main paper. In this sense, I found the way SEINT was presented in the Appendix very helpful. Indeed, on page 22, we see that SEINT is a GW variation where the second transposed coupling matrix is replaced with a product of the two convoluted embeddings depending on it. The text makes the introduction of this metric overly complicated, at least for me. Once we see it through this lens, a natural question is why SEINT outperforms GW? GW is theoretically a SE(p) metric too. It would be great to see some intuitions for why it fails in shape matching tasks, in addition to the favorable computational complexity of SEINT.
2.	To the best of my understanding, the metric is defined as an inf sup problem, yet it is solved algorithmically as a sup inf problem. This can be seen from Algorithm 1 (we solve inf inside the loop and then take max over the solutions). Shouldn’t we use minimax (von Neumann or Sion theorem) somewhere to justify doing this? In general, we know that sup inf <= inf sup, so we cannot use the majorization-minimization argument either, unless f is convex/concave in the first and second argument, which is unlikely given how the problem is defined.
3.	The authors omit some very important GW variations in their literature survey. One such omitted reference that comes to my mind is the Quantized GW (Chowdhury et al.), which is extremely computationally efficient and was proposed, among other applications, for shape matching. Another one is the Sampled Gromov-Wasserstein paper (Kerdoncuff et al.). Its complexity is quadratic in N, and it approximates GW.

**Questions:**

1.	What is the intuition of defining PTD? This is a crucial step in the paper, yet it is not explained very well. Why do we use the cost function defined as the difference between the norm of a vector in X and a real value of the reference measure?
2.	Why are other more efficient GW variants not considered in the evaluations? I can move my score to 6 if a reasonable justification/additional experiments are provided here.
3.	I would like to understand better the algorithmic implementation in the sense of my sup inf remark above. Can authors elaborate more on this?
4.	I had to go to the appendix (page 22) to see how the reference measure \mu_Z is defined. They seem to be uniform measures over the hypercube. Is there any motivation for this?

---

> ### Author Response · Authors · 2025-11-21
> **Response to Reviewer 4D2S (Part 1 / 2)**
>
> We thank the reviewer for acknowledging the novelty of our proposed metric and its computational efficiency. We provide a point-by-point response to your concerns below and have incorporated the necessary revisions into the paper. (Changes are highlighted in blue in the updated PDF.)
> - **On W1: Comparison to GW**
>
>     First, we would like to clarify the missing entries in the SE(p)-Invariant experiments. For GW and EGW, the entries marked with ''---'' indicate that results were not reported due to prohibitive computational costs, rather than poor performance. We believe that GW and EGW would likely yield competitive results given sufficient resources. We apologize for any confusion this may have caused and have added an explanatory note in the revised manuscript.
>
>     Second, we emphasize that **SEINT and GW define distinct metric spaces**. We have added an MDS experiment on the horse-gallop dataset in Appendix D.8 (Lines 1571-1595). By computing the distance matrices of the (rotated) horse sequences and extracting the top two features via MDS to plot trajectories, we observed that both SEINT and GW exhibit consistent periodicity, demonstrating their metric properties and rotation invariance. However, they display different patterns where GW traces a rectangle while SEINT traces a polygon. This indicates that they construct distinct metric spaces.
>
> - **On W2 & Q3: The Minimax Problem**
>
>     In our numerical implementation, under the setting of Corollary 2, for a given $k$, the optimization of the reference distribution $Z$ can be viewed as optimizing the support points $z_1, z_2, \dots, z_k$.
>     Since the minimax theorem primarily relies on function convexity, we provide an illustrative proof to demonstrate the validity of our inf-sup exchange.
>     Recalling our discrete objective function (Lines 1297-1308):
>     $$
>     \min _ P \max _ Z \sum_{i = 1}^n\sum _ {j = 1}^m P_{ij}({C_i^X}^\top\zeta_Z^X - {C_j^Y}^\top\zeta_Z^Y)^2 \\
>     \text{s.t. } P\mathbf{1}_n = p_Y, P^{\top}\mathbf{1}_m = p_X
>     $$
>     It is evident that the objective is linear with respect to $P$. Inspired by the minimax theorems (von Neumann or Sion) mentioned by the reviewer, the exchange is valid if the objective is convex/concave with respect to $Z$.
>     To illustrate, let $m = n$, and assume sorted norms $||x_1||\leq||x_2||\leq...\leq ||x_n||$ and $||y_1||\leq||y_2||\leq...\leq ||y_n||$. Furthermore, let $\mu_Z$ be a discrete uniform distribution on $n$ points $z_1\leq z_2\leq...\leq z_n$. The PTD calculation can be expressed as:
>     $$
>     (\zeta_Z^X)_i = |||x_i|| - z_i|, \quad (\zeta_Z^Y)_i = |||y_i|| - z_i|
>     $$
>     Let $s^X, s^Y$ denote the sign of the absolute value expansion, defined as:
>     $$
>     s^X_i = \operatorname{sign}(||x_i|| - z_i), \quad s^Y_i = \operatorname{sign}(||y_i|| - z_i)
>     $$
>     Let $S^X = \operatorname{diag}(s^X)$ and $S^Y = \operatorname{diag}(s^Y)$. Then we have:
>     $$
>     {C_i^X}^\top\zeta_Z^X - {C_j^Y}^\top\zeta_Z^Y = {C_i^X}^\top S^X(||x|| - z) - {C_j^Y}^\top S^Y(||y|| - z) = ({C_i^X}^\top S^X - {C_j^Y}^\top S^Y)z + \text{const}
>     $$
>     The objective function becomes:
>     $$
>     \sum _ {i = 1}^n \sum _ {j = 1}^n P _ {ij}({C_i^X}^\top\zeta_Z^X - {C_j^Y}^\top\zeta_Z^Y)^2 = \sum _ {i = 1}^n\sum _ {j = 1}^n P _ {ij}(({C_i^X}^\top S^X - {C_j^Y}^\top S^Y)z + \text{const})^2
>     $$
>     It is straightforward to see that $(({C_i^X}^\top S^X - {C_j^Y}^\top S^Y)z + \text{const})^2$ is convex with respect to $z$. Since $P _ {ij} \geq 0$, the weighted sum remains convex.

---

> ### Author Response · Authors · 2025-11-21
> **Response to Reviewer 4D2S (Part 2 / 2)**
>
> - **On W3 \& Q2: Other GW Variants**
>
>     We acknowledge that we overlooked several relevant methods in our original literature survey. We have now revised our Introduction to provide a more comprehensive review of the literature.
>     Furthermore, we recognize that our rotation invariance experiments initially only compared SEINT to a few classical methods. To ensure a more comprehensive comparison, we have tested additional baselines, including: SFGW[1], PGW (as a fast version of SampledGW, [2]), AE[3], QGW[4], and LRGW[5]. The results of this expanded comparison are presented below:
>
>     |Method|Acc (k=1)|Acc (k=5)|Acc (k=10)|Time (ms)|
>     |-|-:|-:|-:|-:|
>     |SFGW|100.00±0.00|100.00±0.00|100.00±0.00|485.185|485.185|
>     |LrGW|2.30±1.94|2.10±1.77|2.00±2.00|1350.36|
>     |PGW|45.60±1.36|42.90±1.83|40.10±1.20|693.47|
>     |QGW|5.00±0.45|5.00±0.45|5.00±0.45|242.50|
>     |AE|2.50±0.89|2.50±0.89|2.50 ± 0.89|15.82|
>     |**SEINT (Ours)**|100.00±0.00|100.00±0.00|100.00±0.00|4.46|
>
>     The table shows that SEINT maintains the highest performance in both speed and accuracy. Notably, among the compared baselines, only SFGW achieves accuracy comparable to SEINT; however, SEINT significantly outperforms SFGW in computational efficiency.
>
>     Furthermore, we have added 3-KNN classification experiments on the Animals and FAUST datasets. The specific results are presented below:
>
>     |distance|animals acc.(%)|animals time(ms)|FAUST-500 acc.(%)|FAUST-500 time(ms)|FAUST-1000 acc.(%)|FAUST-1000 time(ms)|
>     |-|-|-|-|-|-|-|
>     |S(F)GW|99.1±1.2|3.3|36.7±5.8|26.3|40.4±6.0|35.9|
>     |AE|97.9±1.8|1.6|37.8±5.6|13.9|41.6±6.0|15.9|
>     |GW|100.0±0.0|8.3|29.2±4.3|662.5|33.3±5.2|1915.0|
>     |QGW|57.7±5.1|4.8|13.1±3.5|32.6|13.3±3.3|47.7|
>     |PGW|86.6±3.6|133.4|19.1±3.9|695.9|18.6±3.8|1025.9|
>     |**SEINT**|99.3±1.3|0.8|39.3±5.7|5.9|42.3±6.4|19.2|
>     |**ISEINT**|99.4±1.4|0.6|39.4±5.8|4.9|42.9±6.7|13.7|
>
>     The results presented in the table demonstrate that SEINT and ISEINT consistently achieve superior performance.
>
> - **On Q1: Intuition of PTD**
>
>     Our initial intuition stems from the Monge–Kantorovich depth[6], which projects a high-dimensional distribution onto a uniform ball. Since the ball is rotationally invariant, the corresponding depth and the length of the mapping are also rotationally invariant. This inspired us to extract such features. Given that solving OT in multi-dimensional space is computationally expensive, and considering that the reference measure is a radially symmetric distribution, we utilize the radial length distribution to characterize the process. We extract the mapping length of the 1D OT as our rotation-invariant feature (PTD).
>
> - **On Q2:**  Please refer to our response in **W3**.
> - **On Q3:**  Please refer to our response in **W2**.
> - **On Q4: Reference Measure Selection**
>
>     In our theoretical proof, we rely on specific reference measures. Based on these, we expanded the selection space for the reference measure. When taking the supremum over this space, we obtain a ''least favorable measure,'' which ensures that the specific measure we require is theoretically acceptable. To practically encompass the necessary reference measures, we simply considered a hypercube. Our experiments indicate that this choice is sufficiently effective; consequently, we did not investigate more complex reference spaces.
>
> In summary, we hope that your concerns are resolved by our response, helping you reconsider our paper. Feel free to contact us if there are any other questions.
>
> **Reference**
>
> [1] A novel sliced fused Gromov-Wasserstein distance. 2025
>
> [2] Sampled Gromov Gasserstein. 2021
>
> [3] Fast and robust comparison of probability measures in heterogeneous spaces. 2020
>
> [4] Quantized Gromov-Gasserstein. 2021
>
> [5] Linear-time Gromov Gasserstein distances using low rank couplings and costs. 2022
>
> [6] Monge–Kantorovich depth, quantiles, ranks and signs. 2017

---

> > ### Comment · Reviewer_4D2S · 2025-11-25
> > **Thank you for the provided clarifications**
> >
> > I think the authors' reply is reasonable, and I will raise my score to 6. I think it can be instructive to include the new baselines with their algorithmic complexities in the revised version of the pape,r too.

---

> > > ### Author Response · Authors · 2025-11-25
> > > **Thank you**
> > >
> > > We sincerely thank you for reconsidering our work and updating the score. We will incorporate the time complexity analysis into the comparison table in the next version of the paper.

---

> > > ### Author Response · Authors · 2025-12-01
> > > **PDF Update**
> > >
> > > Dear Reviewer 4D2S,
> > >
> > > We have updated the PDF manuscript and added the algorithmic complexities of the new baselines in Table 7 (Lines 1493–1507). We appreciate your valuable suggestion.

---

### Official Review · Reviewer_mdhA · 2025-11-01

**Soundness:** 3
**Presentation:** 4
**Contribution:** 3
**Rating:** 8
**Confidence:** 4

**Summary:**

This paper introduces SEINT (SE(p)-Invariant Transport), a novel metric for comparing structured data that maintains invariance under Special Euclidean transformations. The authors develop two unsupervised representation techniques—Polar Transport Discrepancy (PTD) and Distance-convoluted PTD (DcPTD)—that generate 1D SE(p)-invariant features through optimal transport. They rigorously prove SEINT satisfies metric properties (identity, symmetry, triangle inequality) on isometry classes. Experiments demonstrate SEINT achieves 100% accuracy in transformation recognition while maintaining computational efficiency, and improves molecule generation validity by 3-4% over EDM baselines. The approach effectively balances theoretical soundness with practical utility across multiple domains.

**Strengths:**

**Theoretical Innovation and Rigor**: The paper makes a significant theoretical contribution by establishing SEINT as a proper metric with formal proofs (Section 2.3, Theorem 1). Unlike prior work that "fail to simultaneously optimize for computational efficiency, rigorous metric properties, and general applicability," SEINT achieves this crucial balance. The authors cleverly bridge optimal transport theory with SE(p)-invariance requirements, providing Corollary 2 that establishes metric properties for discrete distributions under realistic assumptions. This theoretical foundation elevates the work beyond heuristic approaches common in the field.

**Practical Impact and Validation**: The paper demonstrates compelling real-world utility across diverse domains. In molecule generation (Table 5), SEINT regularization improves validity from 93.4% to 96.7% while maintaining atom stability. For point cloud classification (Table 6), it consistently enhances accuracy across multiple datasets (e.g., +0.61% on Scan-Objonly). Notably, SEINT achieves perfect (100%) transformation recognition accuracy while being computationally efficient—outperforming alternatives like SW and SGW that "fall below 60% accuracy." This practical validation across multiple challenging tasks underscores the method's significance.

**Weaknesses:**

**Limited Analysis of the Validity-Uniqueness Trade-off**: While Table 5 shows SEINT improves validity but decreases uniqueness (98.2% → 92.0-93.2%), the authors only briefly mention this trade-off as "acceptable." A more thorough investigation would strengthen the work—specifically, quantifying how much uniqueness degradation is attributable to the SEINT regularization versus other factors, and whether certain molecular properties correlate with this trade-off. Including visual examples of molecules where validity improved at the cost of uniqueness would make this analysis more concrete.

**Insufficient Comparison to Recent SOTA Methods**: The evaluation compares against GW, EGW, RISGW, SW, and SGW, but misses several relevant recent approaches like UniGEM (Feng et al., 2024) mentioned in the introduction. Given the rapid evolution in this field, the authors should benchmark against at least 2-3 additional recent methods (e.g., from ICLR/NeurIPS 2024) to better position SEINT's novelty and performance advantages.



**Ambiguity in Reference Measure Selection**: While Section 2.3 discusses reference measure selection, the practical implications of different choices aren't fully explored. The paper states "the support points for µZ are drawn from [0,M]^k" but doesn't analyze how sensitive results are to k (number of reference points) or distribution type. Figure 5 shows some effects, but a more systematic ablation study would strengthen the method's practical guidance.

While the weaknesses I identified (limited analysis of validity-uniqueness trade-off, insufficient comparison to very recent SOTA methods, and reference measure sensitivity) should be addressed fairly easy none fundamentally undermine the contribution. The authors' argument that the validity-uniqueness trade-off is acceptable (since validity is more critical for chemical plausibility) appears reasonable.

**Questions:**

1. In Corollary 2, you assume "values of the norm ||x_i|| are distinct"—how would SEINT behave when this assumption is violated in practice (e.g., symmetric molecules with identical atom distances)? Could you provide a mathematical bound on the error when norms aren't perfectly distinct?

2. The complexity analysis mentions "n log(n)" scaling, but how does SEINT perform in high-dimensional settings (p > 100)? Have you tested scalability to non-geometric data where p is large but structure exists?

---

> ### Author Response · Authors · 2025-11-21
> **Response to Reviewer mdhA (Part 1 / 2)**
>
> We are grateful for the reviewer's strong endorsement and for highlighting the balance between our theoretical innovation and practical utility. Detailed responses to your comments are provided below, and the manuscript has been revised accordingly. (Changes are highlighted in blue in the updated PDF.)
> - **On W1: Validity-Uniqueness Trade-off.**
>
>     As shown in [1], Validity measures the proportion of chemically valid molecules, whereas Uniqueness quantifies the ratio of non-duplicate molecules after removing identical samples.
>
>     **In practice, the phenomenon in which increased validity leads to decreased uniqueness is prevalent in molecular generation due to the complementary nature of these two metrics.** Methods such as END[2], UniGEM[3], and GEOLDM[4] similarly report this trade-off when optimizing for higher validity. Regarding this phenomenon, we have already explained in the paper (Lines 1419-1424).
>
>     Furthermore, based on the definitions, a molecule must first be valid before its uniqueness can be meaningfully discussed. Therefore, the common practice in literature (including the aforementioned papers) is to treat Validity and Uniqueness as a unified performance metric, often comparing the product $V \times U$.
>
>     As shown in Tables 3 and 4 (Line 469-497), after incorporating SEINT as a regularization term, our method achieves a higher $V \times U$ than the baseline. This further demonstrates the effectiveness of our approach in balancing validity and uniqueness under the commonly adopted evaluation protocol.
>
> - **On W2: Comparison to Recent SOTA Methods.**
>
>     To further validate the effectiveness of SEINT, we have incorporated the following additional recent baselines, including SFGW[5], PGW[6], AE[7], QGW[8], and LRGW[9]. The results are presented below:
>
>     |Method|Acc (k=1)|Acc (k=5)|Acc (k=10)|Time (ms)|
>     |-|-:|-:|-:|-:|
>     |SFGW|100.00±0.00|100.00±0.00|100.00±0.00|485.185|485.185|
>     |LrGW|2.30±1.94|2.10±1.77|2.00±2.00|1350.36|
>     |PGW|45.60±1.36|42.90±1.83|40.10±1.20|693.47|
>     |QGW|5.00±0.45|5.00±0.45|5.00±0.45|242.50|
>     |AE|2.50±0.89|2.50±0.89|2.50 ± 0.89|15.82|
>     |**SEINT (Ours)**|100.00±0.00|100.00±0.00|100.00±0.00|4.46|
>
>     The table confirms that SEINT maintains the highest performance in both speed and accuracy. Notably, among the newly incorporated baselines, only SFGW achieves accuracy comparable to SEINT; however, SEINT significantly outperforms SFGW in computational efficiency.
>
>     Furthermore, we have added 3-KNN classification experiments on the Animals and FAUST datasets. The specific results are presented below:
>
>     |distance|animals acc.(%)|animals time(ms)|FAUST-500 acc.(%)|FAUST-500 time(ms)|FAUST-1000 acc.(%)|FAUST-1000 time(ms)|
>     |-|-|-|-|-|-|-|
>     |S(F)GW|99.1±1.2|3.3|36.7±5.8|26.3|40.4±6.0|35.9|
>     |AE|97.9±1.8|1.6|37.8±5.6|13.9|41.6±6.0|15.9|
>     |GW|100.0±0.0|8.3|29.2±4.3|662.5|33.3±5.2|1915.0|
>     |QGW|57.7±5.1|4.8|13.1±3.5|32.6|13.3±3.3|47.7|
>     |PGW|86.6±3.6|133.4|19.1±3.9|695.9|18.6±3.8|1025.9|
>     |**SEINT**|99.3±1.3|0.8|39.3±5.7|5.9|42.3±6.4|19.2|
>     |**ISEINT**|99.4±1.4|0.6|39.4±5.8|4.9|42.9±6.7|13.7|
>
>     In conclusion, SEINT outperforms recent state-of-the-art methods in both experiments. This consistent performance across different tasks strongly validates SEINT's effectiveness.
>
> - **On W3: Ambiguity in Reference Measure Selection.**
>
>     We have added an ablation study analyzing how the scale of $k$ affects the results, which is detailed in the revised manuscript (in Appendix D.7 (Lines 1548-1570)).

---

> ### Author Response · Authors · 2025-11-21
> **Response to Reviewer mdhA (Part 2 / 2)**
>
> - **On Q1: The distinct conditions.**
>
>     First, we clarify the role of this condition. To avoid additional optimization steps for the optimal couplings $\pi^\* _ {X,Z}, \pi^\* _ {Y,Z}$ in Eq. (1), we require $\pi^\* _ {X,Z}, \pi^\* _ {Y,Z}$ to be unique. This introduces the condition that the norms $\|x_i\|$ should be distinct, thereby reducing our implementation cost.
>
>     If the values of the norm $\|x_i\|$ are not distinct, since our algorithm only considers the optimization of $\mu_Z$, the actual optimization space becomes slightly larger than intended. This may lead to results that exceed the ground truth. Unfortunately, we have not yet derived a mathematical bound on this error, but we plan to address this in future work.
>
> - **On Q2: High-dimensional settings.**
>
>     If the dimension $p$ is considered in our computational cost, the complexity of SEINT is $O(n^2+np)$ (where $p$ mainly accounts for the computation of norms), and our accelerated method is $O(n \log n + np)$ (where $p$ mainly accounts for the computation of DcPTDs). It can be seen that our algorithm is linear with respect to $p$, and the acceleration is most effective when $n \gg p$. When $p$ is large, although the computational burden increases slightly, our experimental results in Figure 4(b) (Lines 432-443) and the additional experiment in Appendix D.6 (Lines 1515-1546) demonstrate that our algorithm maintains robust performance in high-dimensional settings.
>
>     Regarding non-geometric data: if our understanding is correct, the ''non-geometric data'' you mentioned refers to topological data. However, our method primarily focuses on data associated with rotation-invariant groups. It appears that general topological data might be outside the scope of this paper. Could you please provide more specific information regarding the data structure of the non-geometric data, or provide some examples? This would help us better determine whether SEINT is applicable to such data.
>
> In summary, we hope the above response successfully addresses your concerns and strengthens your confidence in supporting our work during the discussion and decision phases. Thank you for your consideration.
>
> **Reference**
>
> [1] Masked graph modeling for molecule generation. 2021
>
> [2] Equivariant neural diffusion for molecule generation. 2024
>
> [3] UniGEM: A Unified Approach to Generation and Property Prediction for Molecules. 2025
>
> [4] Geometric latent diffusion models for 3d molecule generation. 2023
>
> [5] A novel sliced fused Gromov-Wasserstein distance. 2025
>
> [6] Sampled Gromov Gasserstein. 2021
>
> [7] Fast and robust comparison of probability measures in heterogeneous spaces. 2020
>
> [8] Quantized Gromov-Gasserstein. 2021
>
> [9] Linear-time Gromov Gasserstein distances using low rank couplings and costs. 2022

---

### Official Review · Reviewer_omzW · 2025-11-01

**Soundness:** 2
**Presentation:** 2
**Contribution:** 2
**Rating:** 4
**Confidence:** 3

**Summary:**

The paper proposes SEINT as a Special Euclidean group-invariant distance between probability measures supported on normed vector spaces. The method builds 1D, rotation/translation-invariant features in two steps: (i) Polar Transport Discrepancy (PTD), which couples the norm distribution of data with a 1D reference measure; (ii) a Distance-convoluted PTD (DcPTD) that computes the convolution of pairwise distances against PTD to encode intrinsic geometry. The SEINT distance is then defined as the optimal transport (OT) distance between DcPTD representations, with a min-max structure that adversarially chooses a “least favorable” reference measure. Theoretically, SEINT is a proved to be a metric on isometry classes, which respects translational and rotational invariances. A discrete variant of SEINT provides metric guarantees under distinct-norm and support-size assumptions. Algorithmically, given sample size $n$, the proposed implementation with a single reference measure achieves $O(n^2)$ computational complexity in general and $O(n \log n)$ when the ground distance admits additional structure (leveraging structure specific matrix decompsoition technqiues for accelerated computation). Empirically, the paper reports perfect rotation/translation robustness on a benchmark 3D point cloud classification task, cross-space (2D - 3D) comparisons, favorable perfomance scaling with dimension, and improved 3D molecule generation based on SEINT regularization.

**Strengths:**

The paper proposes a rotation invariant metric capable of comparing distributions across different ambient dimensions, with theoretical guarantees that establish the metric properties and rotation invariance of SEINT.

**Weaknesses:**

1. Although the paper claims both rotatioanal and translational invariances as its target invariance choices, the definition of SEINT as well as the applications of SEINT in experiment rely on explicit centering of the data points in the source and target domains, so that both source and target has their respective origins as the center of their corresponding distributions. The definition of PTD in Equation 2 implicitly assumes this, sincethe norm is centered around 0, and not around any arbitrary constant. Therefore, it seems that the claimed translation invariance of SEINT follows as a consequence if this origin-based centering approach, and not as a derived property that holds in greater generality. The paper should either clarify whether the definition of SEINT still works without origin-based centering, or restrict the claims to only rotational invariance.
2. The discussion at the beginning of Section 2.3 (Lines 222-241) is dense and not well-written. Given the importance of the role of the reference measure, it is worth improving the presentation of this section of the text and being clear about the existence and selection fo the reference measure.

**Questions:**

Please see the Weakness section, particularly the point about the translation invariance of SEINT.

---

> ### Author Response · Authors · 2025-11-21
> **Response to Reviewer omzW**
>
> We thank the reviewer for acknowledging the rotation invariance of our proposed metric and for providing such valuable comments. We have responded to your specific points below and updated the manuscript to reflect these changes. (Changes are highlighted in blue in the updated PDF.)
> - **On W1: Translational Invariance**
>
>     First, regarding the Euclidean space setting, it is indeed true that when distributions are centered before comparison, any method can achieve translational invariance by pre-centering the target and source distributions.
>     However, we wish to clarify that **we do not strictly enforce ''translational invariance'' by merely centering the two distributions at the very beginning of SEINT.** In our definition, we require ''a set of matched points'' to serve as our ''Base Point'' (as mentioned in Appendix B.3 (Lines 1170-1178)).
>
>     Suppose $x$ and $y$ are two Base Points in metric spaces $(X, d_X)$ and $(Y, d_Y)$, respectively. If $(X, d_X)$ and $(Y, d_Y)$ are isometric with isomorphism $f$, then $f(x) = y$. This implies that **the Base Point carries prior information regarding the isometric mapping, which is equivalent to knowing beforehand that the point pair $x, y$ is matched.** Therefore, the selection of Base Points is crucial.
>     The intuition stems from the concept of a barycenter in a metric space. The barycenters of two mm-spaces $(X, d_X, \mu_X)$ and $(Y, d_Y, \mu_Y)$ are derived as:
>     $$
>     \mathcal{B}_X = \operatorname{argmin} _ {x \in X} \int_x d_X^2(x, x')  d\mu_X(x') , \quad \mathcal{B}_Y = \operatorname{argmin} _ {y \in Y} \int_y d_Y^2(y, y') \, d\mu_Y(y').
>     $$
>    We observe that if $(X, d_X)$ and $(Y, d_Y)$ are isometric with isomorphism $f$, then for $\forall x \in X$, assuming $y = f(x)$, we have:
>     $$
>     \int_y d_Y^2(y, y') \, d\mu_X(y')
>     = \int_y d_Y^2(f(x), f(x')) \, d\mu_Y(y')
>     = \int_x d_X^2(x, x') \, d\mu_X(x')
>     $$
>    Then, we can derive that:
>     $$
>         \min _ {x\in X} \int_x d_X^2(x, x') \, d\mu_X(x')  = \min _ {y\in Y} \int_y d_Y^2(y, y') \, d\mu_Y(y').
>     $$
>     Therefore, we have:
>     $$
>     \int_x d_X^2(\mathcal{B}_X, x') \, d\mu_X(x')
>         = \int_y d_Y^2(f(\mathcal{B}_X), y') \, d\mu_Y(y')= \min _ {y\in Y} \int_y d_Y^2(y, y') \, d\mu_Y(y')
>     $$
>     This implies $\mathcal{B} _ Y = f(\mathcal{B} _ X)$, indicating that the barycenter in a metric space is indeed a suitable Base Point.
>
>     Looking back to Euclidean space settings, minimizing the barycenter objective can be derived by solving its gradient function:
>     $$
>     \frac{\partial}{\partial x}\left(\int_x \|x - x'\|_2^2 \, d\mu_X(x')  \right) = 0,
>     $$
>     The solution is:
>     $$
>     x^\star = \int_x x'\, d\mu_X(x').
>     $$
>     This justifies our use of the mean as the Base Point. Although a direct centering operation (subtracting the mean) can also achieve translational invariance in Euclidean space, our approach is theoretically grounded in the general property of metric space barycenters.
>
> - **On W2: Writing Details**
>
>     We have updated the description in Section 2.3 (Lines 234-253). Briefly, we aim to utilize Eq.(5) and constraint (6) to link the optimal couplings $\pi^\* _ {X,Z}, \pi^\* _ {Y,Z}$ in Eq.(1) with $\mu_Z$, thereby avoiding the simultaneous optimization of $\pi^\* _ {X,Z}$ and $\pi^\* _ {Y,Z}$.
>
> In summary, we hope the above response addresses your concerns and assists you in re-evaluating our work. Please feel free to contact us if you have any further questions.

---

### Author Response · Authors · 2025-12-01
**Summary of the Rebuttal and Discussion**

Dear ACs, SACs, and PCs,

We sincerely thank you for handling our paper, and we greatly appreciate the reviewers' positive feedback on our work. Given the length of the discussion session, we have summarized the key points below for your convenience.
- **Summary of Our Response**

  During the discussion phase, we have addressed the reviewers' questions point-by-point and supplemented our work with additional experiments. Key updates include:
    1) Clarifying the intuition behind translational invariance and the extension of SEINT to general metric spaces.
    2) Demonstrating the convexity of the algorithm implementation and the validity of the minimax exchangeability.
    3) Comparing SEINT against other recent efficient metric methods, including S(F)GW, PGW, QGW, LrGW, and AE. The results confirm that SEINT remains the fastest method while maintaining superior performance.
    4) Adding extensive benchmark experiments, such as MDS visualization of horse sequences, non-normal high-dimensional experiments, and classification experiments based on geodesic distance (Animals & FAUST). SEINT consistently achieves efficient and robust performance across these tasks.
    5) Correcting typos and refining details as pointed out by the reviewers.

  Overall, we believe these concerns have been effectively resolved and should no longer serve as grounds for rejection.
- **Summary of Reviewer Feedback**

  We are pleased to highlight that following the rebuttal process, our scores have been raised to **4, 8, 6, 8, 6** (as of 25 Nov 2025, 00:36, AOE). The detailed feedback is as follows:

     1) **Reviewer omzW** acknowledged the rotation invariance of our method but raised questions regarding translational invariance and readability of Sec. 2.3. In response, we demonstrated that SEINT can be generalized to more general metric spaces and explained that the Barycenter possesses invariant properties (with a simplified form in Euclidean space). We also updated Sec. 2.3 to enhance readability.
     2) **Reviewer mdhA** **highly appreciated the theoretical and experimental contributions of our method**. The reviewer raised questions regarding the trade-off in molecule generation metrics and theoretical conditions, and suggested additional experiments on high-dimensional data and ablations on the number of reference points. We answered these questions point-by-point, compared our method with other recent efficient metrics, and added the requested experiments.
     3) **Reviewer 4D2S** acknowledged the novelty, efficiency, and theoretical grounding of our method.The reviewer questioned the minimax exchangeability and several technical details, and suggested comparisons with PGW and QGW. We provided a detailed explanation of the convexity in our numerical experiments, addressed the remaining questions, and included comparisons with recent methods.. Following our response, **Reviewer 4D2S stated that our answers were reasonable and raised their score to 6 (25 Nov 2025, 00:36, AOE).**
     4) **Reviewer SoMY** highly recognized our work, stating it is "clearly written, original, and results are strong." The reviewer offered some minor suggestions, which we have addressed. Following our response, **Reviewer SoMY confirmed that our work is very solid and maintained their positive score (26 Nov 2025, 06:56, AOE).**
     5) **Reviewer dTTa** praised our work from multiple perspectives. The reviewer raised questions regarding presentation, theoretical results, and numerical experiments, specifically asking about potential extensions. We addressed the presentation and theoretical queries and performed comprehensive comparative experiments, including those requested by the reviewer and an extended fused version. Following our response, **Reviewer dTTa stated that their concerns about the experiments were mainly addressed and raised their score (24 Nov 2025, 00:58, AOE).**

  In summary, our work has received an overall positive assessment from Reviewers mdhA, 4D2S, SoMY, dTTa. Reviewer mdhA assigned a high overall score in the initial review, and Reviewers 4D2S, SoMY, dTTa explicitly stated that their concerns were resolved during the discussion. While Reviewer omzW has not provided further feedback, we carefully responded to all points raised in the initial review and believe that these concerns have been clarified.

In light of this, we respectfully hope that the converging positive views of Reviewers mdhA, 4D2S, SoMY, dTTa will be taken into account as evidence of the contribution and soundness of our work. We also kindly request that Reviewer omzW's initial concerns be considered together with our detailed rebuttal, in order to assess whether they have been sufficiently addressed. We are grateful for the time and effort you and the reviewers have invested in evaluating our submission, and we thank you again for coordinating the review process and for your fair and careful assessment.

Best regards,

Authors (Submission ID: 22539)

---

### Meta-Review · Area_Chair_YabW · 2026-01-07

**Summary:**

The paper presents a novel appraoch for comparing probability distributions in measured Banach spaces, which is based on the extraction of invariant representations. The reviewers generally agree about the novelty and the merits of the prposed approach. While some concerns were raised about translation invariance, clarity or comparisons with more recent baselines, most of the reviewers concur that the authors replies were succesful in addressing them. For these reasons, I am inclined to give an accept option for this paper.

**Reviewer Concerns:**

see general comment

**Reviewer Scores:**

dTTa indicated that he raised his score.

---

### Decision · Program_Chairs · 2026-01-26

Accept (Poster)